# Evaluating the effect of stand properties and site conditions on the forest reflectance from Sentinel-2 time series

**Ewa Grabska**[1,2]*, **Jarosław Socha**[2]

**1** Institute of Geography and Spatial Management, Faculty of Geography and Geology, Jagiellonian University, Kraków, Poland, **2** Department of Forest Resources Management, Faculty of Forestry, University of Agriculture in Kraków, Kraków, Poland

* ewa2.grabska@doctoral.uj.edu.pl

**Data Availability Statement:** The data used in this study are not owned by the authors and can be accessed from the following third party sources: The Sentinel-2 datasets analyzed during the current study are available in the Copernicus Open Access

## Abstract

Forest stand reflectance at the canopy level results from various factors, such as vegetation chemical properties, leaf morphology, canopy structure, and tree sizes. These factors are dependent on the species, age, and health statuses of trees, as well as the site conditions. Sentinel-2 imagery with the high spatial, spectral, and temporal resolution, has enabled analysis of the relationships between vegetation properties and their spectral responses at large spatial scales. A comprehensive study of these relationships is needed to understand the drivers of vegetation spectral patterns and is essential from the point of view of remote sensing data interpretation. Our study aimed to quantify the site and forest parameters affecting forest stands reflectance. The analysis was conducted for common beech-, silver fir- and Scots pine-dominated stands in a mountainous area of the Polish Carpathians. The effect of stands and site properties on reflectance in different parts of the growing season was captured using the dense time series provided by Sentinel-2 from 2018–2019. The results indicate that the reflectance of common beech stands is mainly influenced by elevation, particularly during spring and autumn. Other factors influencing beech stand reflectance include the share of the broadleaved understory, aspect, and, during summer, the age of stands. The reflectance of coniferous species, i.e., Scots pine and silver fir, is mainly influenced by the age and stand properties, namely the crown closure and stand density. The age is a primary driver for silver fir stands reflectance changes, while the stand properties have a large impact on Scots pine stands reflectance. Also, the understory influences Scots pine stands reflectance, while there appears to be no impact on silver fir stands. The influence of the abovementioned factors is highly diverse, depending on the used band and time of the season.

## Introduction

The spectral properties of vegetation depend on a variety of factors, such as the leaf water content, pigment and non-pigment leaf constituents, and internal leaf structure [1, 2]. Forest

Hub (https://scihub.copernicus.eu/). The Sentinel-2 Level-2A product was used in this study. Four Sentinel-2 tiles were used: 34UFA, 34UFV, 34UEA, and 34UFV. The Forest Data Bank datasets (Forest inspectorates data) are available in shapefile formats from their website (https://www.bdl.lasy. gov.pl/portal/wniosek-en). Digital Elevation Model from ISOK project is available to download here (https://mapy.geoportal.gov.pl/).

**Funding:** This research was supported by the project I-MAESTRO. Project Innovative forest MAnagEment STrategies for a Resilient biOeconomy under climate change and disturbances (I-MAESTRO) is supported under the umbrella of ForestValue ERA-NET Cofund by the National Science Centre, Poland and French Ministry of Agriculture, Agrifood, and Forestry; French Ministry of Higher Education, Research and Innovation, German Federal Ministry of Food and Agriculture (BMEL) via Agency for Renewable Resources (FNR), Slovenian Ministry of Education, Science and Sport (MIZS). ForestValue has received funding from the European Union's Horizon 2020 research and innovation programme under grant agreement N˚ 773324. The APC was funded by the Ministry of Science and Higher Education of the Republic of Poland for the University of Agriculture in Krakow for 2020.

**Competing interests:** The authors have declared that no competing interests exist.

reflectance at the canopy level, i.e., as seen from satellite imagery, is a complex phenomenon resulting from the leaf chemical properties, morphology, distribution, canopy structure and various tree sizes [1, 3]. Furthermore, at the stand level, a spectrum can represent a mixture of materials with different reflectance properties [4]. These characteristics are dependent on the tree species, age, and health status, and additionally the forest structure properties, such as the canopy closure, as well as the site conditions, e.g. elevation [5–8]. Finally, the reflectance is related to the spatial resolution of the used imagery, where very high-resolution data allow individual crowns to be distinguished, while with lower spatial resolutions, pixels represent a mixture of different canopies and often the background.

The role of individual sources of vegetation reflectance variability is controlled by the scattering and absorption of electromagnetic waves of different lengths [1]. The typical reflectance of healthy vegetation is characterized by a low reflectance in the visible range, excluding green waves, an abrupt increase in the red-edge region (~700 nm; RE), high reflectance in the near-infrared (NIR) region, and a decrease around 1500 nm in the short-wave infrared (SWIR) part of the spectrum [2]. In visible waves, reflectance largely depends on pigment content [9]. In the RE region, there is a transition from strong absorption by chlorophyll to high reflectance in the NIR region, related to the scattering of internal leaf structures and the canopy [9–11]. The SWIR part of the spectrum is sensitive to the leaf and canopy water content [10, 12]. The reflectance of vegetation changes during the growing season, corresponding to the phenological cycle. Reflectance variability is more pronounced in deciduous species; however, evergreen conifers also exhibit seasonal reflectance variations [8].

Regarding the stand age, changes in the canopy reflectance are related to vegetation physiological and structural properties [13]. In general, younger trees tend to have higher reflectance than older ones [14]. With an increasing tree age, a decline in foliage photosynthesis occurs [15]. In the case of conifer trees, the leaf area index (LAI) is correlated with the age of trees [16], and needle age has an impact on near-infrared transmittance and reflectance [17, 18]. In the case of broad-leaved forests, the influence of forest age on reflectance has also been reported in the near-infrared region [14]. However, the differentiation of stand age classes is rather a result of structural differences, for example, in very young forest stands, the reflectance may be dominated by the understory vegetation or soils [19]. On the other hand, in older stands, the roughness of the forest canopy increases, there is a lower density of stems [20], and gaps are more numerous and larger, which causes the increase of shadowing and, therefore, lower stand reflectance [14, 21]. Furthermore, there is an effect of the understory vegetation on stand reflectance [22], and it changes during the growing season [23]. Particularly, with open crown closure, more understory vegetation can be seen from remote sensing data [24], and the contribution in sparse canopies can be even larger from the understory than the tree canopy reflectance itself [25]. Thus, at the stand level, the influence of the background signal may differ in stands with similar species compositions but differing canopy closures or crown shapes [3, 26]. The abovementioned relationships have been used in studies for predicting and classifying forest stand age using optical remote sensing data [16, 27–32] and determining the successional stages of forests [33, 34].

Other factors which have an impact on vegetation reflectance include the site conditions. Different illumination and light environments can affect the optical properties of leaves and needles [6]. In the case of deciduous species, there is a large phenological variability caused by elevation-related temperature gradients and the local microclimate [35–37]. Typically, there is a negative correlation between leaf onset and elevation [36]. In remote sensing studies, the impact of elevation changes on reflectance values has been reported for paper birch [38] and spruce [39]. Furthermore, trees elevation-related stress has been reported [38, 40]. Trees disturbances can significantly influence stand reflectance, causing, for example, leaf discoloration

and defoliation [41]. The reflectance differences between the same tree species may also occur due to site fertility [42, 43], which can also influence forest floor reflectance [44, 45]. Still, the influence of all the aspects mentioned above on forest reflectance has not yet been fully explored at the stand level, as for example, that which is seen with high-resolution satellite imagery.

With satellite imagery, it is now possible to study the vegetation properties and site conditions impact on reflectance at larger scales and with higher frequency. The high temporal resolution is essential, as the vegetation changes can be very rapid, particularly at the beginning and the end of the growing season. In recent years, Sentinel-2 imagery has been successfully used in determining various forest stand properties. With 10 and 20 meter spatial resolutions, Sentinel-2 imagery allows assessment of at both the tree- and stand-level scales [8]. The repetition cycle of Sentinel-2 mission is two to five days. Furthermore, Sentinel-2 sensors provide 13 spectral bands–particularly SWIR and RE bands are promising in analysis of vegetation properties. In studying relationships between growing stock and volume, particularly Sentinel-2 SWIR1 band was characterized as highly correlated [46]. Similarly, the role of three narrow RE bands should be examined, as, for example, the RE1 band was found as the most important in predicting forest structure parameters [47].

Although the general principles of forest stand reflectance are well studied, the detailed information on reflectance variability patterns in space and time is still needed. Particularly, the multivariate analysis, including different site and stand properties as predictors is essential, as commonly these drivers are considered separately. These relationships should be studied for different tree species, both deciduous and ever-green. In general, this information is valuable to interpret the remote sensing signals, analyze spectral patterns of vegetation and understand the relevant drivers [1, 18, 48]. Also, there is a need to understand how the different stand properties influence the reflectance in terms of estimating forest variables from satellite imagery [26]. Thanks to this knowledge, satellite imagery can be used to determine the features of stands, such as species composition, density, growing stock volume and biomass or the age of stands [49–51]. Understanding the impact of forest stand properties and site conditions on the forest reflectance is of crucial importance for applying remote sensing technologies to monitor forest ecosystems. Linking the diversity of forest reflectance caused by different stands and site characteristics allows for applying the satellite remote sensing in forest inventory. The research results may also be of importance in the assessment of forest sites. Particularly when analyzing large geographic extents with various conditions, e.g. characterized by an extensive range of elevations, and forest stand properties, these factors should be considered.

This study aims to evaluate how the different forest stand parameters and site conditions influence the forest stand reflectance of common beech-, silver fir- and Scots pine-dominated stands in mountainous areas, specifically the Polish Carpathians. The dense time series of Sentinel-2 are used to provide a comprehensive analysis showing species and site-specific spectral responses in different parts of the growing season. The research novelty consists of the simultaneous consideration of the impact of site conditions, the structure of stands, the age of trees and understory vegetation on broadleaved and coniferous forests reflectance. In particular, the influence of the following parameters on forest stand reflectance are examined:

- Site conditions: elevation, aspect, and slope;

- Forest structure: stand density and crown closure;

- Tree age;

- Understory vegetation type.

## Materials and methods

### Study area

For the analysis, a test site of approximately 13,000 km$^2$ was selected. It is composed of 28 Forest Districts (i.e. a basic forest management unit in the Polish State Forests structure), located in southeastern Poland (Fig 1). These Forests Districts were selected based on two criteria: 1) laying in the Carpathians, the Carpathians foothills or the Outer Subcarpathia region and, 2) laying within the area of four examined Sentinel-2 tiles. The study site consists of mountainous areas (Lesiste and Mid-Beskidy Mountains), as well as mountain foothills (Mid-Beskidy Foothills) and basins (Sandomierz Basin). At the test site, elevation ranges from 150 to 1180 meters a.s.l., and forest covers approximately 38% of the area, predominantly distributed in the mountainous areas. The dominant forest species are common beech (*Fagus sylvatica*), silver fir (*Abies alba*) [52], and Scots pine in the foothills and basins (*Pinus sylvestris*); [53]), and these are the three species analyzed in this study. Forest management systems applied in the study area are adjusted to the forest species composition. Beech-dominated stands are managed with shelterwood system; silver fir dominated stands are managed by stepwise cutting, whereas Scots pine stands are mostly managed with a clear-cutting system.

### Satellite imagery collection and pre-processing

Multi-temporal Sentinel-2 imagery from the years 2018 and 2019 were used in this study (Fig 2). Although the "base" year for our analysis was 2019, we used three Sentinel-2 images from

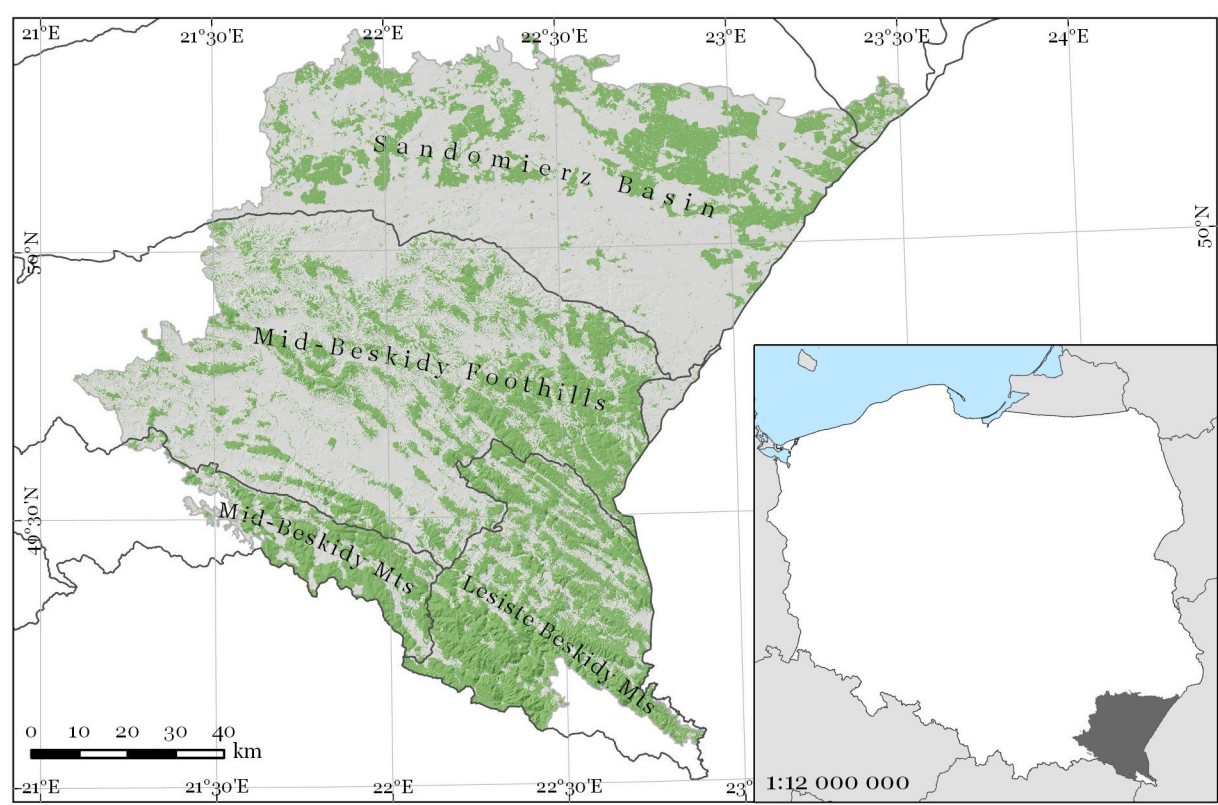

**Fig 1. Study area located in southeastern Poland.** Green areas represent forest cover. Forest cover mask was freely downloaded from Copernicus Land Monitoring Service (https://land.copernicus.eu/pan-european/high-resolution-layers/forests).

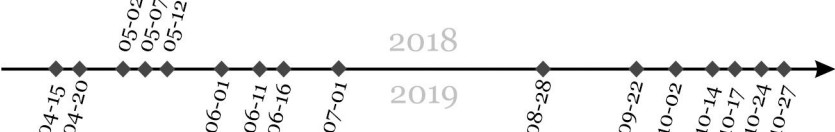

**Fig 2. Sentinel-2 acquisition dates in years 2018 and 2019.**

May 2018, as data from this part of the season was missing in 2019 due to heavy cloud cover. Sentinel-2 Bottom-of-Atmosphere products were downloaded using the sen2r package in R [54]. The study site contained four Sentinel-2 tiles, (34UFA, 34UFV, 34UEA, and 34UFV). Ten bands with 10- and 20-meter spatial resolutions were used, i.e., visible blue, green, and red, RE 1–3, NIR 1–2, and SWIR 1–2. All images with cloud cover less than 30% were downloaded, and then they were further visually inspected for clouds. Finally, 16 dates were selected for analysis. For the selected images, cloud, cloud shadow, and snow masking were applied. These masks were derived from the Sentinel Land Cover classification product [55], and the masking was performed in R using raster package [56].

## Reference data

Reference data about the forest stand species were derived from the Forest Data Bank (FDB) for the year of 2019. The FDB is a freely available dataset containing information on state forests in Poland. It consists of so-called subareas (stands), i.e., homogeneous forest areas for which the share of particular species is known. The information used for our study included tree species share, tree age, crown closure, stand density, and understory vegetation. For the analysis of reflectance, only stands with a 100% share of a particular tree species were selected. Here, 12,235 stands were selected, including 2726 stands for common beech, 2201 for silver fir, and 7308 for Scots pine. The mean size of the analyzed stands was 8.6 ha. The age of the analyzed beech stands ranged from 4 to 174 (mean of 87.5), fir from 5 to 162 (mean of 82.8), and pine from 2 to 151 years (mean of 68.6). For each stand, the information on stand properties was assigned from FDB, i.e., the stand age, crown closure, stand density, and understory (species and share).

A microclimate is strongly related to topography, which characterizes climate conditions [57]. Therefore, in order to characterize site conditions as the indirect measures of regional climate variations, we used elevation above sea level. Local microclimate variation was also characterized with aspect and slope. Topographic characteristics were calculated using a 1-meter resolution Digital Elevation Model (DEM) acquired from the Polish IT System for Country's Protection Against Extreme Hazards. The DEM was resampled to a 10-meter resolution, and the slope and aspect were calculated (Table 1). Then, the mean values of the environmental variables were assigned for each stand polygon.

Finally, for further analysis, the mean reflectance values from each Sentinel-2 band for all examined dates were extracted to stand polygons.

## Regression models

Firstly, in order to determine which properties have an impact on stand reflectance we performed preliminary analysis. This was carried out with correlation matrices and the visual analysis of scatterplots. Furthermore, a cleaning function was applied to the data to remove outliers arising from erroneous observations. Multivariate outliers were detected using the minimum covariance determinant [58] using the MASS package [59].

**Table 1. Predictive variables used in this study.**

| | Variables | Description and units |
|---|---|---|
| **Site conditions** | Elevation | Meters above sea level |
| | Slope | Degrees |
| | Aspect | Degrees |
| **Stand properties** | Age | Years |
| | Crown closure | 1 (open) |
| | | 2 (sparse) |
| | | 3 (moderate) |
| | | 4 (close) |
| | Stand density | Determined by comparing the stand volume (wood volume) actually existing in the stand of a given species per 1 hectare to the potential volume, given by comparing the yield tables of a fully stocked stand of a given site index. For example, if the actual volume of the stand is 240 m$^3$/ha while 300 m$^3$/ha is shown in the yield tables, then the stand density is 240/300 = 0.80. Rarely has values above 1. |
| | Understory vegetation | Coniferous/broadleaved: 0 (no understorey) to 10 (full understory) |
| | | Additionally, in the case of Scots pine stands, for the purpose of visualization, the information on the dominant understory species was used (beech, oak, fir). |

Based on the preliminary analysis step, using the most important variables for the selected dates and bands, regression models were fitted. Mean stand reflectance values from ten analyzed Setninel-2 bands were used as dependent variables. We used Generalized Additive Models (GAMs; [60]), where the dependent variable is modeled as a sum of various smoothing functions. GAMs are flexible and easy to interpret, and they can model complex relationships [61, 62]. The performances of the models were assessed using the adjusted coefficient of determination (R$^2$). Both single and multiple regression models were constructed (Table 2). GAMs were fitted in R using mgcv package [61].

# Results

## Common beech

The elevation was the most important driver for beech stand reflectance changes, particularly during spring (the beginning of May) and autumn (the middle of October; Table 3; Fig 3). The elevation is also more significant than two other environmental variables (Fig 5).

**Table 2. Single and multiple GAMs fitted in this study.**

| species | Predictor variables | dates |
|---|---|---|
| **Beech** | Elevation | Spring, autumn |
| | Age | Summer |
| | Understory vegetation | Spring, autumn |
| | Elevation + slope + aspect | Spring, autumn |
| **Fir** | Age | All year |
| | Age + stand density + slope | All year |
| **Pine** | Age | All year |
| | Stand density | All year |
| | Understory | All year |
| | Age + stand density + elevation + understory | All year |

**Table 3. Selected adjusted R² values for the GAMs regression for three examined species.**

| Species | Predictors | Date | Band | Adjusted $R^2$ |
|---------|-----------|------|------|----------------|
| **Beech** | Elevation | May 2$^{nd}$ | Blue | 0.57 |
| | | | NIR1 | 0.50 |
| | | May 12$^{th}$ | Green | 0.40 |
| | | | RE1 | 0.44 |
| | | | NIR1 | 0.55 |
| | Age | July 1$^{st}$ | NIR1 | 0.33 |
| | Understory | April 15$^{th}$ | SWIR1 | 0.28 (all stands) 0.55 (stand density < 0.5) |
| | | October 27$^{th}$ | SWIR1 | 0.22 (all stands) 0.41 (stand density < 0.5) |
| | Elevation + slope + aspect | May 12$^{th}$ | RE1 | 0.6 |
| | | October 17$^{th}$ | NIR1 | 0.56 |
| **Fir** | Age | April 20$^{th}$ | NIR1 | 0.58 |
| | | September 22th | RE2-RE3 | 0.53 |
| | Age + stand density + slope | April 20$^{th}$ | NIR1 | 0.61 |
| | | May 12$^{th}$ | Green | 0.57 |
| **Pine** | Age | July 1$^{st}$ | SWIR1 | 0.17 |
| | Stand density | July 1$^{st}$ | NIR1 | 0.29 |
| | Understory | April 20$^{th}$ | SWIR1 | 0.40 (stand density < 0.5) |
| | Age + stand density + elevation + understory | July 1$^{st}$ | NIR1 | 0.49 |
| | | April 20$^{th}$ | SWIR1 | 0.50 |

In general, the patterns and spectral behavior for the region from RE2 to NIR2 were very similar. In spring, the relationships were positive between the visible and RE1 reflectance and elevation while negative for RE2-NIR2 region (Fig 3). The strongest relationships between the elevation and reflectance of beech stands occurred for visible and RE1 reflectance on all May dates ($R^2$ = 0.4–0.5). For RE2-NIR2, the strongest correlations were observed on May 2$^{nd}$ and 7$^{th}$. The changes in reflectance depending on elevation were very dynamic in the first half of May (Fig 3). On May 2$^{nd}$, the most significant differences in reflectance were observed above an elevation of approximately 700 m a.s.l. On May 7$^{th}$, there was an increase in green and RE1 reflectance, reaching a maximum at an elevation of approximately 900 m a.s.l., then starting to decrease. On May 12$^{th}$, RE2-NIR2 did not show strong relationships with elevation. Also, there were no clear patterns in terms of reflectance-elevation dependence in the summer months. During autumn, the strongest relationships occurred in the middle of October, and most of them were negative (Fig 3). Afterwards, in the second part of October, again, no clear relationships between beech stand reflectance and elevation were found.

During summer, i.e., from June to September, the influence of the elevation was negligible, and the effect of stand age on the reflectance became more apparent. Younger stands had higher reflectance, and the strongest impact of age was observed in the RE2-NIR2 region, with maximum $R^2$ values achieved in the NIR1 band on July 1$^{st}$ (0.33).

There were also differences between beech stands with different understory vegetation conditions (Fig 4). In general, lower reflectance characterized stands with a lower share of the broadleaved understory. The strongest influence here was found in the SWIR part of the spectrum, and it was also observed in red and RE1 bands in early spring (April) and autumn. The influence of the understory was observed particularly in stands with a sparse crown closure and lower stand density, where the highest $R^2$ values between broadleaved understory and

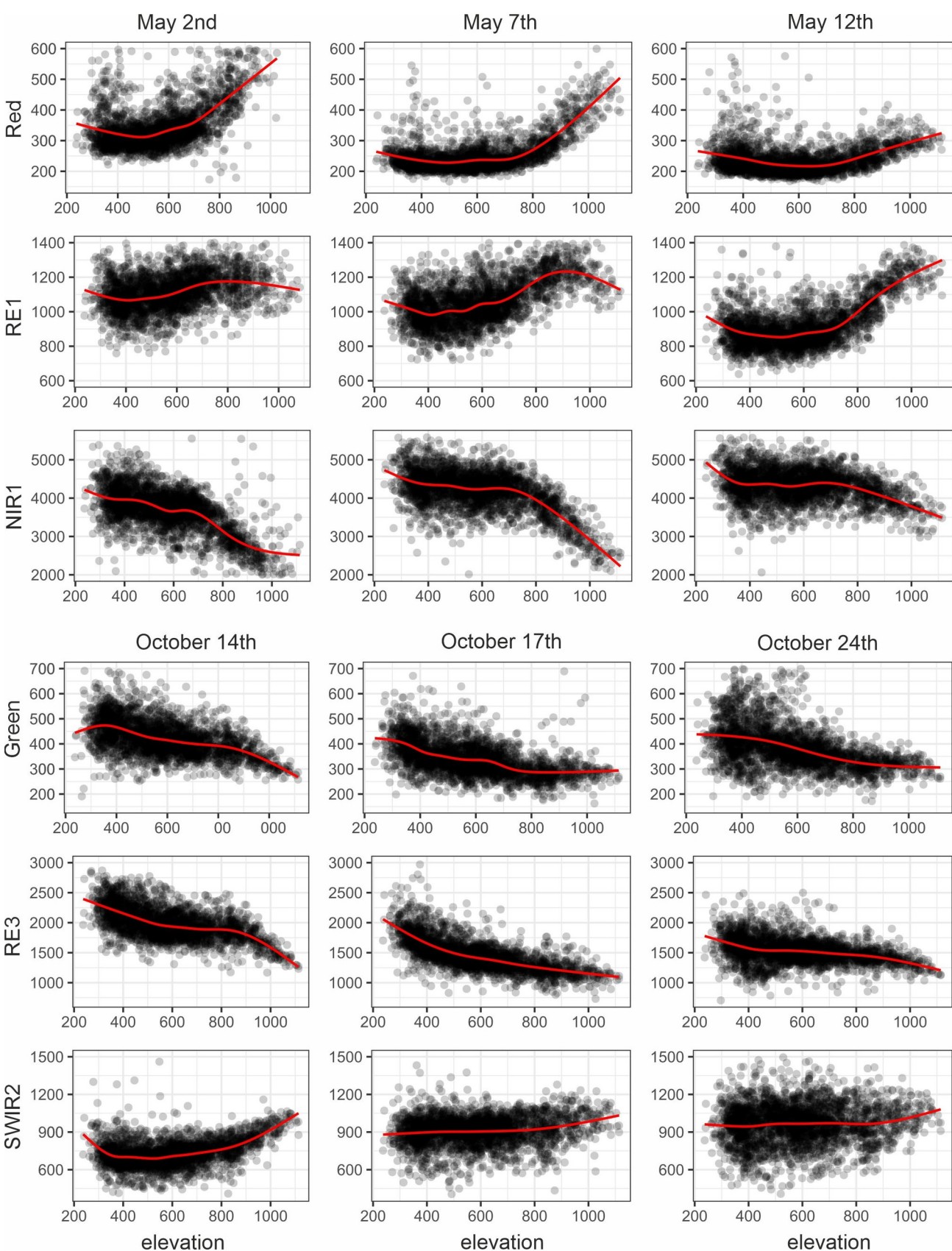

**Fig 3. Common beech-dominated stand reflectance depending on elevation in meters above sea level in selected bands during spring/autumn.**

reflectance were 0.55 for SWIR1 on April 15th (when only stands with a density less than 0.5 were considered), 0.32 (stand density < 1), and 0.28 (all stands). During autumn, the maximum values of $R^2$ were observed on October 27th, specifically 0.41 (stand density < 0.5), 0.28 (stand density <1), and 0.22 (all stands), also in SWIR1 band. Weaker relationships were also observed for RE1 and visible red bands during spring and autumn.

Other environmental variables with a smaller impact on beech stand reflectance were the aspect and slope. The best GAMs fit with multiple predictors was found for RE1 reflectance as a function of the elevation, aspect, and slope on May 12th with an adjusted $R^2$ value of 0.60 (Fig 5). The stands with southeastern expositions (100–150˚) tended to have lower reflectance than northwestern ones. During autumn, the best GAMs were obtained for the RE2-NIR2 region. For NIR1, on October 17th, the $R^2$ value equaled 0.56, also with the elevation, aspect, and slope as predictive variables (Fig 5).

## Silver fir

In silver fir stands, trees age had the highest impact on reflectance, particularly in the RE and NIR bands (Table 3). All the coefficients between age and reflectance were negative here— higher reflectance characterized younger stands. The highest values of $R^2$ were found in April, September (Fig 6), and October, with a maximum of 0.58 on April 20th. However, in general, $R^2$ exceeded 0.4 in RE2-NIR2 part of the spectrum for almost all of the examined dates. It can be noticed that the mean reflectance of stands is entirely separable in classes up to 50 years old in the RE2-NIR2 range (Fig 6).

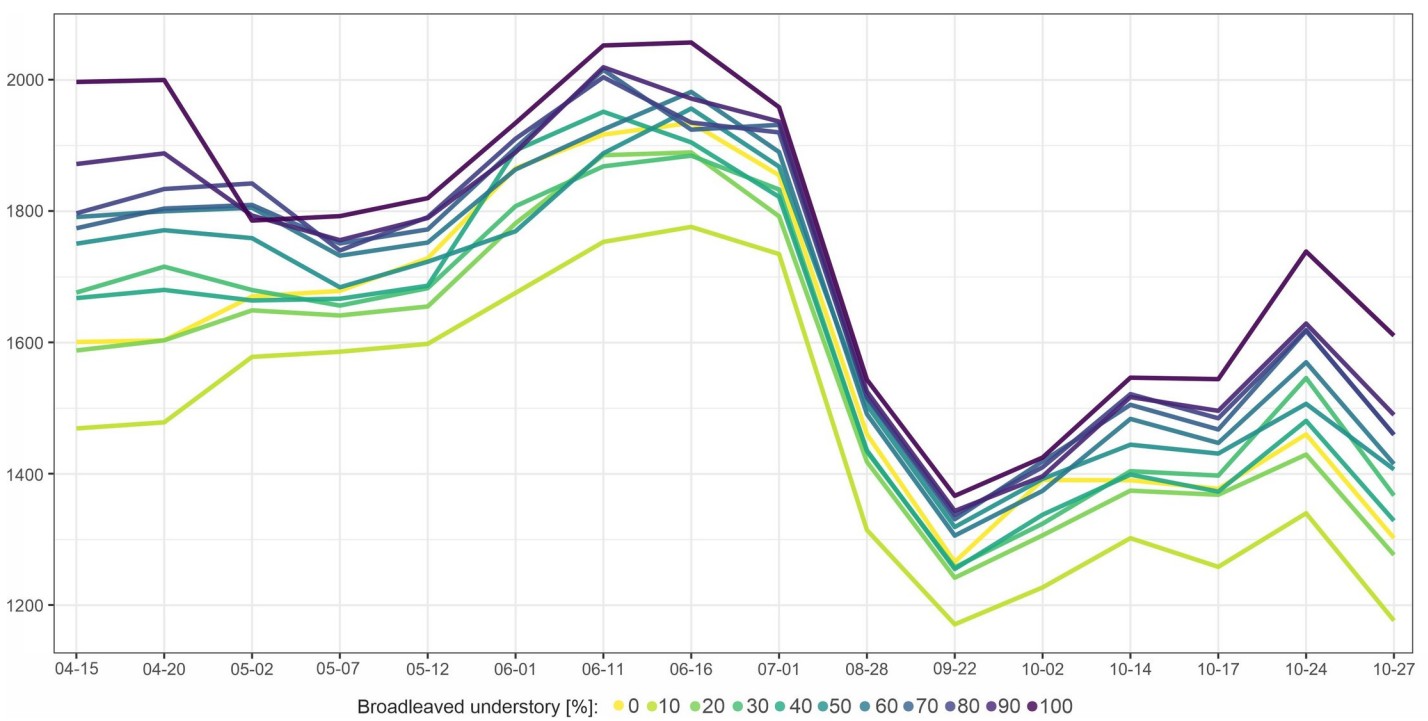

**Fig 4. Beech stands reflectance in SWIR1 band depending on broadleaved understory share.**

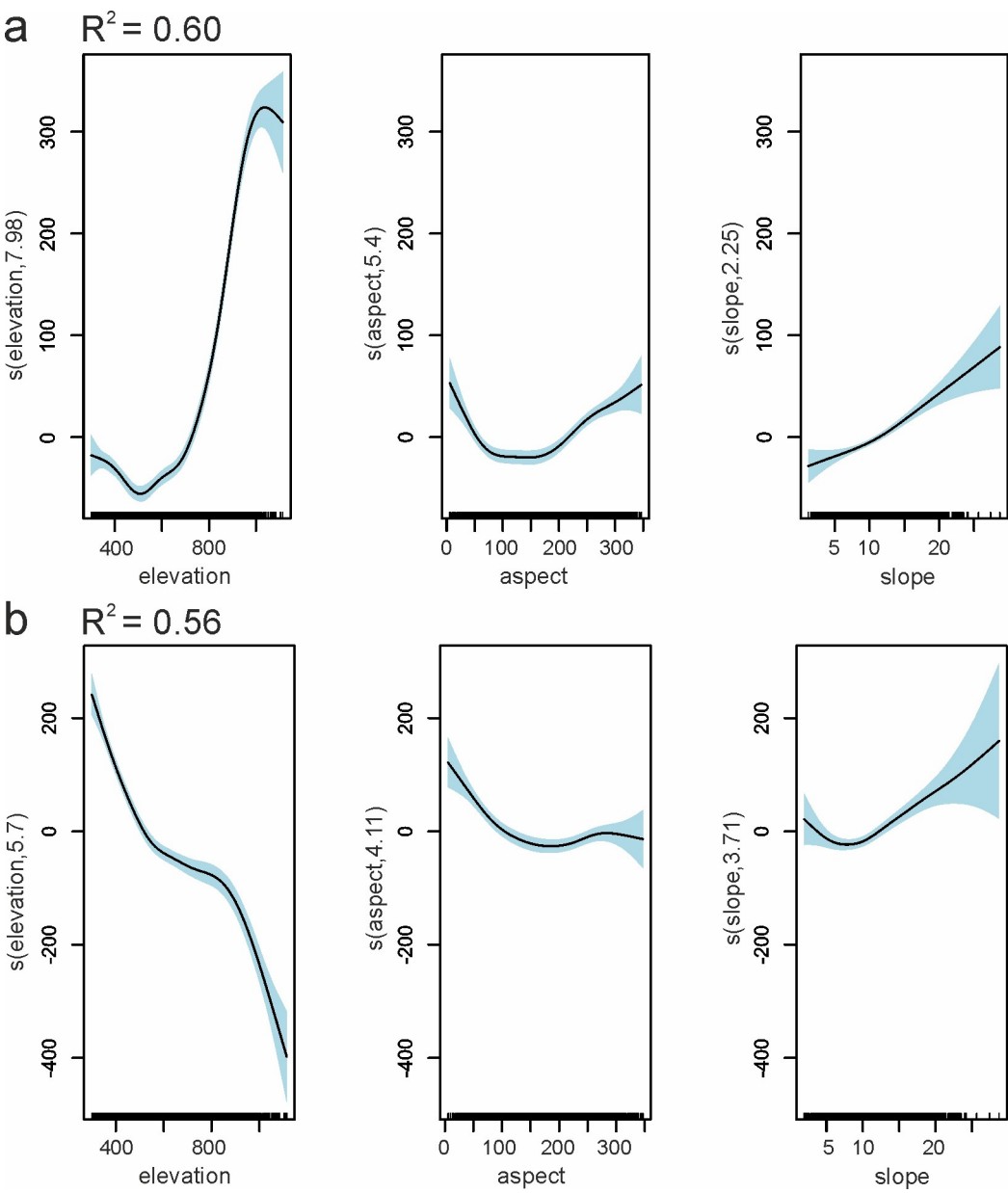

**Fig 5.** GAMs results–visualization of partial effects on common beech stands reflectance from predictors: a) RE1 on May 12th; b) NIR1 on October 17th. The solid line represents the relationship between predictor and response variables, and the light blue shaded area represents confidence intervals.

The structural properties, i.e., stand density, also had an impact on stand reflectance (Fig 7); however, the influence is significantly lower than that of age. Importantly, these two variables are correlated with each other, with correlation coefficients equal to -0.64 (age and stand density). The GAMs, which were developed using many predictor variables, confirm the highest impact of stand age, but also that stand density and slope slightly influenced the reflectance (Fig 7). These relationships were similar throughout the year. Effect of stand density on the reflectance was visible up to a density of approximately 0.6–0.7. Observations with a density above 1 were sporadic; therefore, there were no clear patterns after exceeding this threshold. In the case of the slope, there was a slight increase in reflectance with an increasing slope.

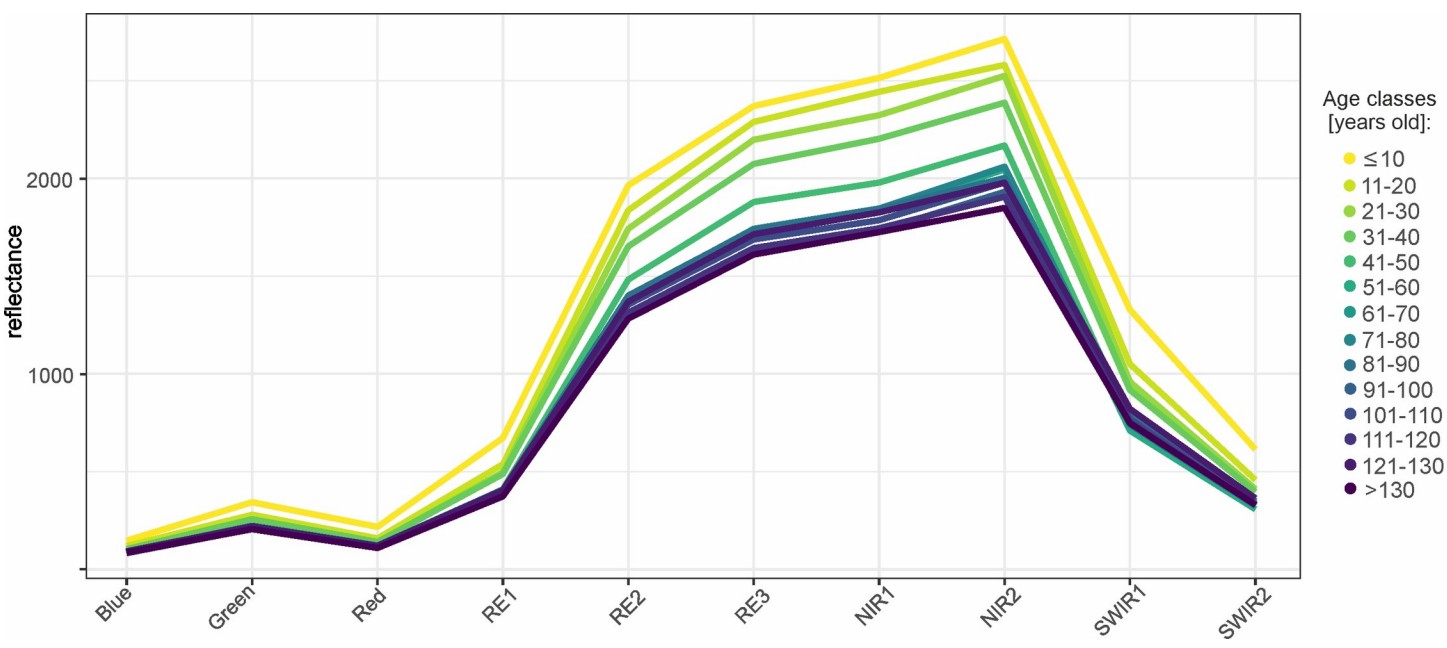

**Fig 6. Mean reflectance for the 10-years age classes of Silver fir stands on September 22nd.**

## Scots pine

The impact of age on Scots pine stand reflectance was much lower than in the case of silver fir stands (Table 3). The strongest influence was observed in the SWIR1 band during spring and

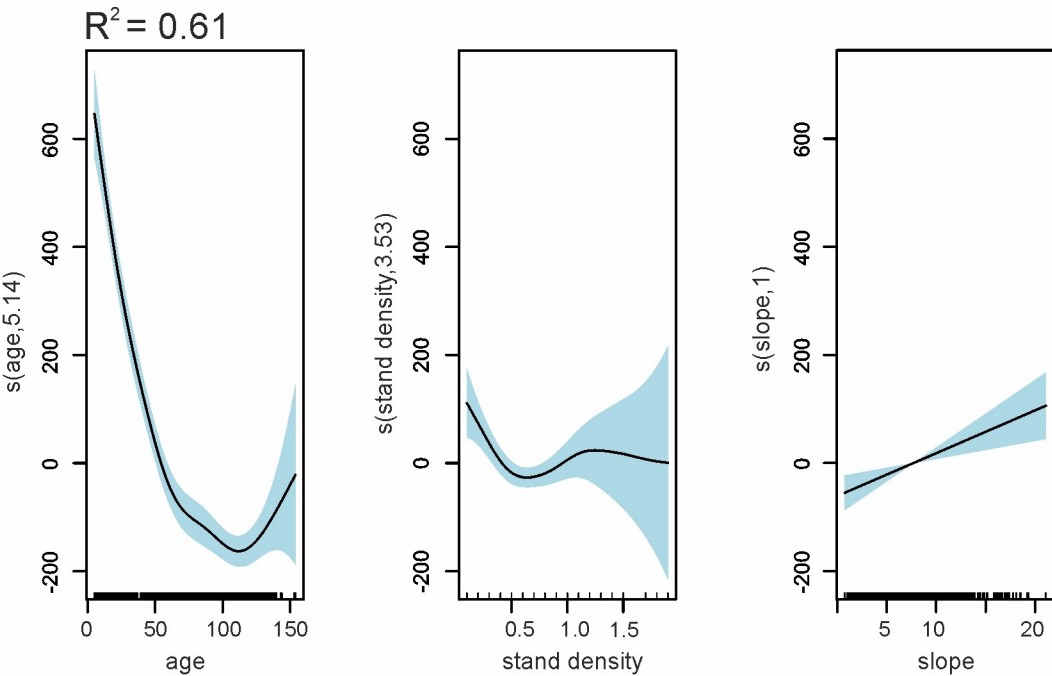

**Fig 7. GAMs results–visualization of partial effects on silver fir stands reflectance from predictors—reflectance in NIR1 on April 20th.** The solid line represents the relationship between predictor and response variables, and the light blue shaded area represents confidence intervals.

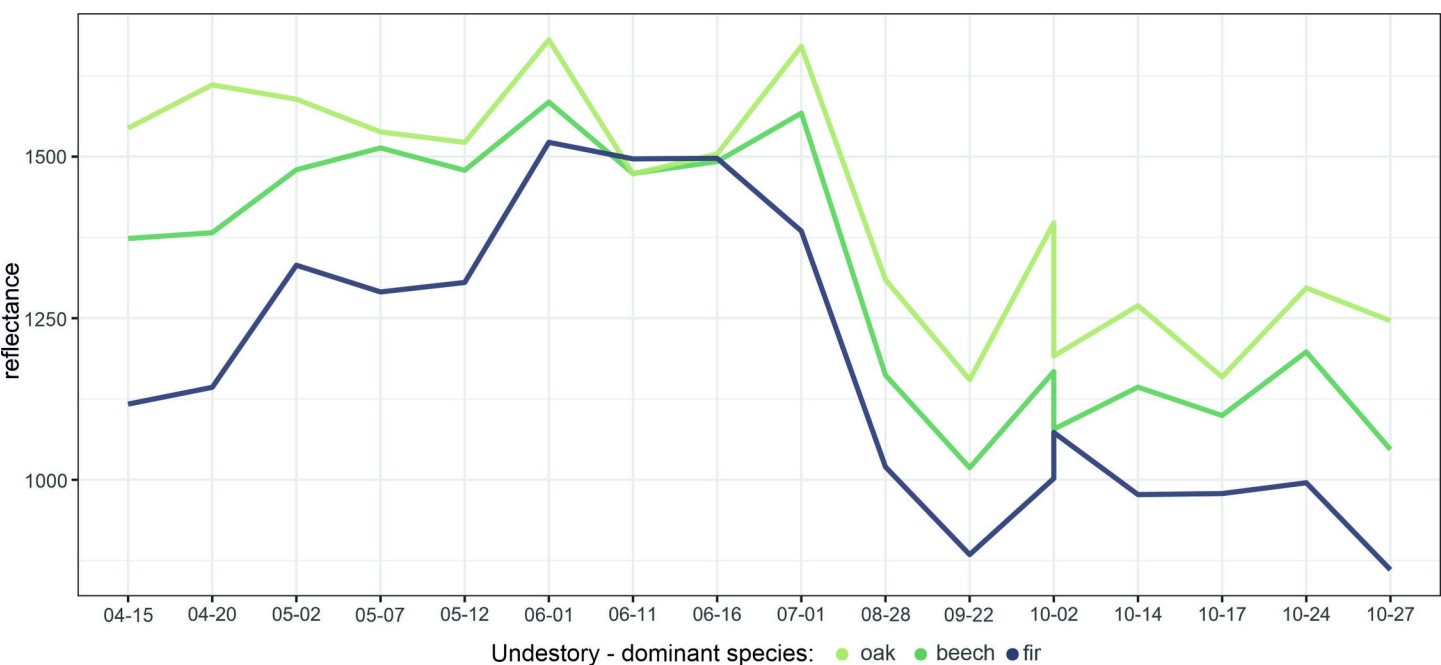

**Fig 8. Scots pine stands reflectance in SWIR1 band depending on dominant understory type.**

summer with a maximum $R^2$ value of only 0.17 achieved on July 1st. The structural parameters, such as crown closure and stand density, were more influential on this species reflectance. Similarly, as in the case of silver fir stands, these variables were correlated with each other. The correlation coefficient between the stand density and crown closure was approximately 0.55, and the correlation coefficient between crown closure and age was -0.47. The impact of the forest stand properties on pine stands was the highest in the SWIR1 (almost all growing season), NIR1 (late spring, summer), and red and RE1 (early spring, late autumn) bands. The strongest relationship between stand density and reflectance occurred during summer in the RE2-NIR2 part of the spectrum with the maximum values observed in NIR1, where the $R^2$ was 0.29. The relationships in all bands were negative, i.e., with higher stand density, reflectance decreased.

The relationships between the broadleaved understory share and SWIR1 were relatively strong, but depending on the crown closure and stand density. In only open crown closure stands, $R^2$ was equal to 0.40, and for stands with a density below 0.5, $R^2 = 0.40$. The understory influence can also be observed in the spectral curves (Fig 8). In April, there were the highest differences in the reflectance depending on the understory, particularly in the SWIR, red, and RE1 bands. Furthermore, the difference between beech and oak understories was noticeable, with slightly higher reflectance from the latter.

Using the GAMs with multiple predictor variables (i.e. stand density, broadleaved understory share, elevation and age), the achieved adjusted $R^2$ values were nearly 0.5 (Fig 9).

## Discussion

In this study, the influence of stand properties and site conditions on forest stand reflectance was analyzed. It was evaluated for stands with the dominance of three species, specifically common beech, silver fir, and Scots pine. The main driving factors for the stand reflectance changes were elevation (common beech), age (silver fir), and stand density (Scots pine). The influence of particular factors notably changed during the growing season.

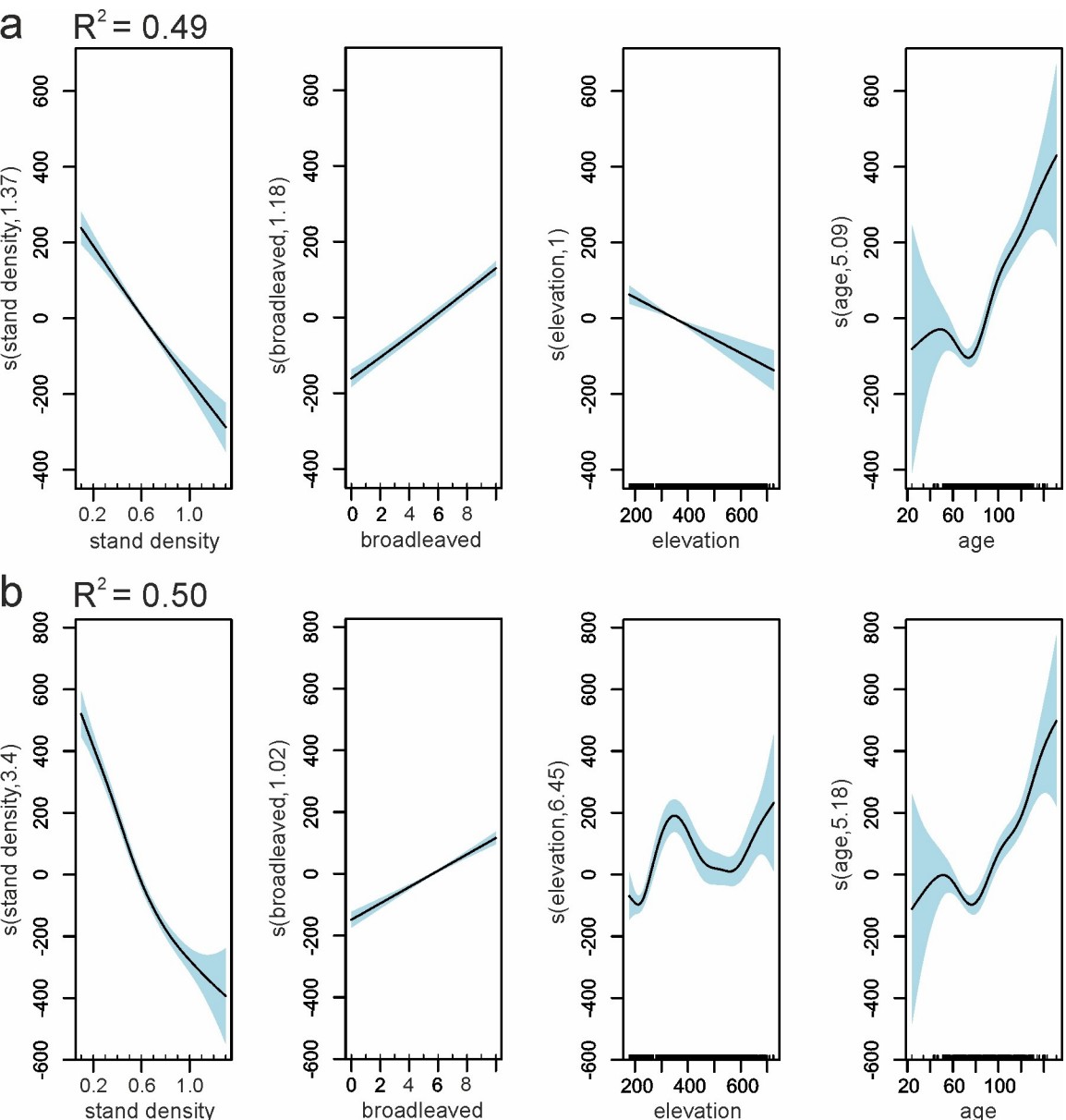

**Fig 9.** GAMs results–visualization of partial effects on Scots pine reflectance from predictors: a) NIR1 on July 1st; b) SWIR1 on April 20th. The solid line represents the relationship between predictor and response variables, and the light blue shaded area represents confidence intervals.

### Common beech reflectance related to site conditions

Due to the phenological phases, elevation was the most important variable affecting beech-dominated stand reflectance during spring and autumn. Visual inspection of the images used in this study allows determining the approximate dates of spring phenophases (Fig 10). In the second half of April, leaves were not present yet. On May 2nd, unfolded leaves were present up to approximately 800–900 meters a.s.l, on May 7th, up to 900–1000 meters a.s.l., and on May 12th, in the entire studied area (i.e., up to 1200 meters a.s.l.). This negative relationship between elevation and the date of leaf unfolding has been reported in previous studies [63, 64], and a high diversity of phenological phases along elevation gradients has been highlighted [65].

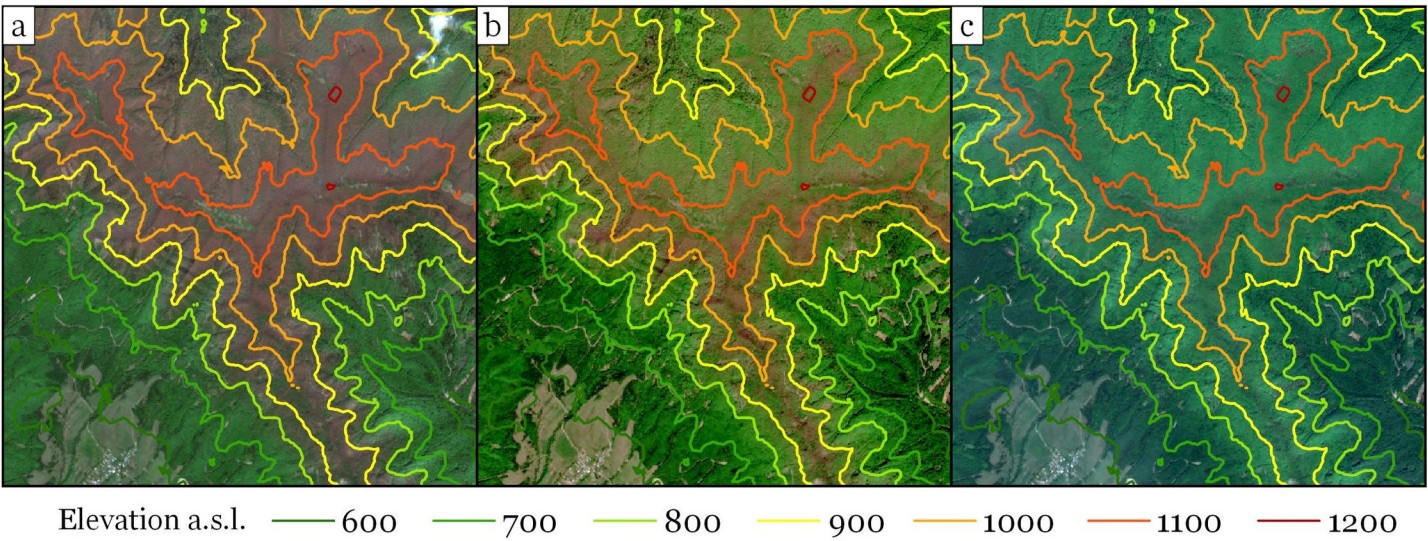

Elevation a.s.l. —— 600 —— 700 —— 800 —— 900 —— 1000 —— 1100 —— 1200

**Fig 10.** Leaf unfolding in beech stands seen in Sentinel-2 images: a) May 2nd; b) May 7th; c) May 12th. The Sentinel-2 imagery was freely downloaded from Copernicus Open Access Hub (https://scihub.copernicus.eu/).

There was a positive correlation between beech reflectance and elevation in the visible and RE1 bands during the spring, while negative in the RE2 to NIR2 wavelengths (Fig 3). It follows that stands with more mature leaves have a lower reflectance in the visible and RE1 regions and higher reflectance in the RE2-NIR2 regions than stands with younger leaves. On May 2nd, the differences in reflectance related to elevation were the strongest in the visible blue ($R^2$ = 0.57) and red bands ($R^2$ = 0.38). The increase of reflectance with increasing elevation can be attributed partially to the LAI, which is larger at lower elevations. A similar trend was reported by Hallik et al. [14] in hemiboreal forests, where a negative relationship between LAI and reflectance was reported in visible bands, particularly in blue and red, and by Eklundh et al. [66] in boreal coniferous forests, where the red band has been found as the most important band for LAI estimation. Also, at a leaf scale, full leaf onset (young leaves) is characterized by a higher reflectance than final leaf onset (mature leaves) in the visible green and red wave regions [67], which can also be observed in our canopy scale analysis. As chlorophyll strongly absorbs energy in the red region [2], more of the red light is absorbed with increasing chlorophyll content at lower elevations. Also, Rautiainen et al. [43] reported LAI and chlorophyll content as the driving factors for changes in the green and red bands.

Patterns similar to green reflectance were observed in the RE1 band, suggesting that similar drivers cause changes in both of these bands. On May 2nd, in the visible green and RE1 bands, the reflectance increased slightly up to approximately 750–800 m a.s.l. and then decreased. This was the approximate border of leaf unfolding here. Similarly, on May 7th, green and RE1 reflectance increased to approximately 900 m a.s.l., and R2 was 0.37 for RE1 and 0.30 for green. The decrease in reflectance in these bands after reaching the maximum cannot be explained either by LAI or by the canopy chlorophyll content, as no leaves were present yet at these higher elevations. Therefore, one of the reasons may be the presence or properties of the understory vegetation. A few days later (May 12th), when the leaves had already unfolded throughout the whole studied area, the peak reflectance in these bands moved to the maximum elevations and the relationships were stronger, with $R^2$ achieving 0.44 for RE1 and 0.40 for green. RE1, similarly to the visible bands, has been reported to be sensitive to the chlorophyll content [68]. The part of the spectrum found as the most sensitive to needle chlorophyll

content is located around 715 nm [69]. Also, in boreal forests, the lower reflectance in the RE1 and green bands may be related to higher canopy cover and LAI [70].

Very similar patterns characterized reflectance from the RE2 to NIR2 bands. On May 2nd and May 7th, beech stands at lower elevations had a higher reflectance. RE2-NIR2 reflectance was already very high at these lower elevations and started to decrease above 700–750 m a.s.l. On May 12th, the changes in RE2-NIR2 reflectance with an increasing elevation were considerably smaller. The maximum $R^2$ value was found in the NIR1 band on May 2nd (0.50). With the ageing of the leaves, increased reflectance in the NIR may result from the development of air spaces in the mesophyll of leaves and the presence of reflecting surfaces [2]. The NIR bands are sensitive to LAI variations, and with an increasing LAI, the reflectance in this part of the spectrum is higher [43, 71]. However, the relationship between NIR and LAI has not been reported in the study by Hallik et al. [14].

The autumn beech phenophases exhibited an opposite direction, and the differences across the elevation gradients occurred mainly in the middle of October (Fig 3). In previous studies, the relationships between elevation and autumn phenophases have been reported to be weaker than in the spring [65, 67], or even negligible [72]. The two main phenological phases of broad-leaved trees during autumn are leaf coloring and leaf fall. With leaf senescence, chlorophyll degradation and nitrogen decline occur [73]. Other pigments, including carotenes and xanthophylls, become more evident, and the red anthocyanin synthesis is highlighted [2, 74]. Although the visible bands are sensitive to pigment contents, only a slight relationship was observed in the green band during autumn ($R^2$ of 0.33), and no relationships were observed in the visible red or blue bands. The strongest relationships between reflectance and elevation were found for the RE2, RE3, and NIR1 bands on October 17th, with $R^2$ values exceeding 0.5. At higher elevations, beech stands had a lower reflectance in these bands. One of the reasons for the decrease in NIR reflectance is the reduction of biomass in healthy leaves [75]. On October 17th, a negative relationship was also found in RE1 ($R^2$ of 0.43).

Besides elevation, beech-dominated stand reflectance was slightly dependent on the aspect and slope (Fig 5). Analogous to elevation gradients, the spring phenophases occurred earlier on slopes with a southeastern aspect. The strongest influence of the aspect was noticeable in RE1 in May—stands on slopes with an aspect of 100–150° had lower RE1 reflectance. During autumn, this influence of aspect on reflectance was not straightforward. Regarding slope, in general, stands located on steeper terrain were characterized by a higher reflectance, particularly during spring. The differences in air temperature, as well as the soil fertility and temperature, resulting from the elevation, aspect and slope, may influence the phenophases, physiological processes, and chemical composition of leaves. The consequence of this can be probably observed in differences between stands reflectance. In spring and autumn, the sum of direct solar radiation daily heat on the south-facing slopes is much greater than that of the north-facing slopes [76]. Furthermore, according to Trepińska [77], as a result of the difference in the amount of solar radiation, the top layer of soil on southern slopes is significantly warmer than that of flat ground. Another hypothetical cause for the reflectance variations may be different humidity and the nutritional status of trees and, in particular, leaves. With an increasing elevation, there is higher precipitation, but weaker nutrition, and soils become increasingly deficient in minerals, particularly nitrogen.

## The influence of stand age and forest structure on coniferous species stands reflectance

Stand age had the highest impact on the reflectance of silver fir-dominated stands. In each examined case, i.e., for the entire growing season and in all wavelengths, younger fir stands

were characterized by a higher reflectance than older ones. The strengths of the relationships depend on the given band and time of the year. The strongest relationships occurred during the spring and autumn months in the RE2-NIR2 bands with maximum $R^2$ values above 0.5 obtained in the RE2 (740 nm), RE3 (783 nm), and NIR1 bands (842 nm). This variability is mainly observed in stands to 50–60 years in the RE2-NIR2 range (Fig 6). Similarly, in the study by Hallik et al. [14], the negative correlation of hemiboreal forest age and reflectance has been observed in the NIR region (736–783 nm bands). In general, young fir stands are more homogenous in terms of their tree structure, and the contrast within stands is small; therefore, the reflectance is higher. In younger stands, with trees growing, there is a decrease in the proportion of sunlit ground, and after full canopy closure is achieved, the LAI does not change largely [14]. The slightly stronger relationships in the spring and autumn months may result from the larger internal shadows during this time of the year, and, therefore, the highest impact on reflectance.

In Scots pine stands, the variables that had the most substantial influence on reflectance include stand density and crown closure. Stands with a higher density had lower reflectance, particularly in the summer months and in the NIR region (maximum $R^2 = 0.29$ in NIR1). A smaller impact came from the understory vegetation and age. Interestingly, pine stands with two types of the understories, i.e., oak- and beech-dominated understories, can be distinguished, particularly in the SWIR bands (Fig 8). In contrast to silver fir, for Scots pine, the relationships with age were observed mostly in the visible and SWIR bands, and they were positive; however, with $R^2$ values below 0.2—the youngest stands are characterized by higher NIR reflectance.

The abovementioned differences between the drivers that have the highest impact on silver fir and Scots pine stand reflectance arise from the species characteristics, such as the tree architecture and crown morphology. Also, silver fir is a shade-tolerant, while Scots pine—shade-intolerant species, which determines the stand structure as these species tree growth process is different. Different structural attributes change with stand age in temperate conifer forests, i.e., the mean crown diameter, canopy depth, stem density, and LAI. While the first two increase linearly with time, the last two are non-linear and achieve maximum values at approximately 20 years [20]. Scots pine stands reach the maximum crown coverage at approximately 20–40 years, depending on the initial stand density [78]. In the case of the site conditions in Poland, the full crown closure is probably obtained earlier. The stocking density and crown ratio also decreases with an increasing Scots pine age [78]. Also in our study, there is a slight decrease in Scots pine reflectance observed up until approximately 30 years of age, while, in the case of silver fir, refelctance decreased until approximately 50–60 years of age depending on the band. Similarly, Kuusinen et al. [79] reported the spruce and pine albedos as a decreasing exponential function of age; however, the impact of stand age was weaker for pine than for spruce. There is also a relationship between the basal area and transmittance [80].

In the present study, the strongest relationships between age and reflectance were found in the RE2-NIR1 bands for silver fir and the visible and SWIR bands for Scots pine. In the case of stand density, the SWIR and NIR regions were also important. These bands have also been reported to be valuable in other studies on forest structural properties and age. The strongest relationships between stand age and reflectance have been reported for the SWIR, NIR, and visible red bands [29, 81]. The best predictors of tree density include the visible and SWIR2 Landsat bands [82]. Strong negative correlations between red and NIR reflectance and forest structural properties and age have been reported [83]. Blue, NIR, and SWIR bands have been found to be highly important for predicting the structural factors and age of lodgepole pine trees [84]. Similarly, the blue, red, and NIR bands are strongly related to the stand properties such as crown closure and tree density [85].

Still, even the same species growing in different conditions can be characterized by different, specific tree architectures [86]. For example, Norway spruce, which was not analyzed in our study, has longer branches and higher crown volumes when growing in mixed stands than in the pure ones [87]. The crown morphology of Norway spruce can also adapt to mountainous regions [88]. In the case of Scots pine, there are modifications in the tree architecture observed between ecotypes in Poland [89].

## Understory vegetation influence on stands spectral properties

The impact of understory vegetation on stand reflectance was present both in broad-leaved and coniferous species (Figs 4 and 8). In beech-dominated stands, the understory influence is mainly observed in early spring and autumn, i.e., the time of the year when canopy leaves were partially absent. The differences in reflectance depending on the understory were observed in the SWIR, red and RE1 bands. In Scots pine stands, the impact of the understory, particularly in the SWIR region, was observed during almost the whole season. Unlike beech and pine stands, the understory did not influence silver fir stands, and it may result from the difference in the crown morphology, tree architecture, and the vertical structure of silver fir stands. Other studies have also reported high values in the red and NIR bands for explaining variations caused by the understory [24]. An increase in reflectance during spring can be attributed to understory greening before canopy greening [90, 91]. Furthermore, leaves in the understory can have distinct traits compared to canopy leaves [92]. Also, in some cases, the understory contributes mainly to the total LAI in stands [24, 93]. These effects of the understory can be noticed, for example, in Fig 4, where beech stands with a higher share of broadleaved species had a higher SWIR reflectance during spring.

Interestingly, at a leaf scale, there are also differences between understory and canopy leaf reflectance, and these differences are particularly prominent in the SWIR region [92]. Heiskanen et al. [22] reported the understory drying in coniferous stands from early May to early June. Therefore, seasonal variations in understory spectra may be due to the moisture regime/water content, and this may also be a reason for the high importance of the SWIR band. Also, Doninck et al. [94] reported that understory variability is strongly correlated with Landsat bands (SWIR1, SWIR2, NIR). As highlighted by Rautiainen et al. [8], measurements from SWIR wavelengths are rare for studying forests properties. In future studies on analyzing understory vegetation, this region may be beneficial.

However, it should be kept in mind that only pure and, in case of determining the age impact, even-aged stands were analyzed in this study, while, in fact, most stands are composed of different species, and many are composed of different age classes. For such stands, integrated effects resulting from a variety of structural and environmental factors influence the spectral responses, and the reflectance at particular wavelengths is driven by multiple traits [95]. The stands with mixed species composition and including different age classes should be further analyzed using Sentinel-2 imagery. Another factor that was not analyzed in our study is forest disturbances. Particularly in recent years, large areas of temperate forests have been disturbed due to various reasons, e.g., climate change, more frequent droughts, and bark beetle infestations. In Poland, Norway spruce is particularly at high risk due to dieback and Scots pine stands have also been exposed to disturbances recently. Unhealthy or dead trees are characterized by different spectral responses. Reflectance can also vary depending on the soil moisture and fertility. Using meteorological data and information about the site productivity could potentially contribute to increasing the precision of the models considered here. Therefore, these factors should also be taken into consideration in future studies. Finally, as pointed out by Nilson et al. [25] the role of each contributing factor in forest stand reflectance demands linkage with empirical reflectance as well.

## Conclusions

In this study, Sentinel-2 time series were used in examining the influence of seven different forest and site properties on stand reflectance for three tree species. The results show the complexity of stand reflectance, which differs considerably depending on the used wavelength and the time of the year.

The main driver of beech stands reflectance variability was the elevation, which strongly affects the phenophases, particularly during spring and autumn. In spring, the reflectance relationships with elevation changed dynamically. Firstly, the RE2-NIR2 part was mainly influenced by the elevation, while, a few days later, this impact increased in the green and RE1 bands. Similarly, during autumn, the highest impact of elevation was observed in the RE and NIR bands. Other factors affecting beech stand reflectance included the share of the broadleaved understory, aspect, and, during summer, the age of stands.

On the other hand, the reflectance of two examined conifers showed only a slight influence of the site conditions considered in this study, while more pronounced—of the structural properties and age. These relationships differed between two coniferous species. Similarly, as in the case of beech, the highest impact on reflectance was observed in the RE and NIR bands. The primary driver of changes in the reflectance of the silver fir stands was age, while in Scots pine stands—the stand density. The type of understory also influenced reflectance of Scots pine stands, with no significant impact in case of silver fir stands. In analyzing the understory impact, particularly the SWIR reflectance variability proved to be useful.

## Supporting information

**S1 Table. Adjusted R2 values for the GAMs regression considering the elevation and reflectance of common beech stands during selected dates.**
(DOCX)

**S2 Table. Adjusted R2 values for the GAMs regression considering the age and reflectance of Silver fir stands during selected dates ($> = 80\%$ same-age-dominated fir stands).**
(DOCX)

## Acknowledgments

We thank Mária Potterf, Giovanni Santopuoli, and the anonymous reviewer for their careful revision and valuable suggestions.

## Author Contributions

**Conceptualization:** Ewa Grabska.

**Data curation:** Ewa Grabska.

**Formal analysis:** Ewa Grabska.

**Methodology:** Ewa Grabska, Jarosław Socha.

**Supervision:** Jarosław Socha.

**Visualization:** Ewa Grabska.

**Writing – original draft:** Ewa Grabska, Jarosław Socha.

**Writing – review & editing:** Ewa Grabska, Jarosław Socha.

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
