## [Decision Letter · Decision Letter 0]

21 Dec 2020

PONE-D-20-38051

Evaluating the effect of stand properties and site conditions on the forest reflectance from Sentinel-2 time series

PLOS ONE

Dear Dr. Grabska,

Thank you for submitting your manuscript to PLOS ONE. After careful consideration, we feel that it has merit but does not fully meet PLOS ONE’s publication criteria as it currently stands. Therefore, we invite you to submit a revised version of the manuscript that addresses the points raised during the review process.

All three reviewers feel that your study can be published after major revison. When preparing your revised version, please make sure that all reviewers comments are addressed, specifically:

- make sure that the methodology is clear enough to allow replication (see specific comments provided by the all reviewers

- ensure that all models developed in your study include all necessary details to check for consistency and replicability

We look forward to receiving your revised manuscript.

Kind regards,

Michal Bosela, Ph.D.

Academic Editor

PLOS ONE

Journal Requirements:

2. We note that Figure 1 and 16 in your submission contain map images which may be copyrighted. All PLOS content is published under the Creative Commons Attribution License (CC BY 4.0), which means that the manuscript, images, and Supporting Information files will be freely available online, and any third party is permitted to access, download, copy, distribute, and use these materials in any way, even commercially, with proper attribution. For these reasons, we cannot publish previously copyrighted maps or satellite images created using proprietary data, such as Google software (Google Maps, Street View, and Earth). For more information, see our copyright guidelines: http://journals.plos.org/plosone/s/licenses-and-copyright.

2.1.    You may seek permission from the original copyright holder of Figure 1 and 16 to publish the content specifically under the CC BY 4.0 license. 

Reviewers' comments:

Reviewer's Responses to Questions

**Comments to the Author**

1. Is the manuscript technically sound, and do the data support the conclusions?

Reviewer #1: Yes

Reviewer #2: Partly

Reviewer #3: Yes

2. Has the statistical analysis been performed appropriately and rigorously? 

Reviewer #1: Yes

Reviewer #2: N/A

Reviewer #3: Yes

3. Have the authors made all data underlying the findings in their manuscript fully available?

Reviewer #1: Yes

Reviewer #2: No

Reviewer #3: Yes

4. Is the manuscript presented in an intelligible fashion and written in standard English?

Reviewer #1: Yes

Reviewer #2: Yes

Reviewer #3: Yes

5. Review Comments to the Author

Reviewer #1: General comment:

The manuscript is interesting and well done. However, I think that 17 figures are too much for a scientific publication. I suggest to report some data in the table format, particularly those data about the comparison among variables (e.g. aspect, slope and elevation) for each tree species. The introduction is well written, while material and method could be improved. In particular, the description of study area could be improved and the description of the selected images, and the pre-processing details could be enriched.

The result section is a bit confused, due to the abundance of figures. I suggest to reorganize the text to follow a linear description starting with the comparison among variables and then focusing on the variables that resulted more impacting on the tree species reflectance.

The discussion is fine, even if I suggest to enrich the section with the text reported in the conclusion (Lines 499-515), because the conclusion, in my opinion, should be more concise and focused on the findings obtained.

Finally, please check for the use of acronyms in the whole text.

Specific comments

L39: “openness of trees” do you mean “tree density” or “canopy gaps/closure”?

L61: please use the acronym since you already introduced it in the line 47.

L55-74: very nice introduction. However, I suggest to add few words for the uneven-aged stands. In this text you focused on the even-aged stands describing how the reflectance differs between young and old trees, and between broadleaved and conifer species. Is it possible to discriminate the reflectance of uneven-aged and mixed forest stands through remote sensing data?

L118-120: please use the same style to introduce the tree species and for references.

L114-120: how did you select the 13000 km2? Is it included in an administrative region or a different administrative border? It is not clear which is the criterion used to select the study area. If possible, please add more information about forest ownership, forest management system, stand age and other information useful to describe the forest stands.

L123: Could you explain why only 3 images were downloaded for the 2018?

L128-129: this is the second time that you introduced the acronyms RE, NIR, and SWIR. Please check in the instructions for the authors about the use of acronyms.

L155: can you better explain the masking procedures (i.e. the R package used). Is it again sen2r?

L158: “12,235 samples were selected” from where? Are they randomly generated by the authors? Are they national forest inventory samples? Could you describe better how you select the samples?

L177: you stated that “elevation was the most important driver for beech stand…”. Is it possible to show the differences respect to other site conditions (i.e. slope, aspect)? Otherwise I can’t see this statement. Maybe you could use here the text of lines 200-206 and figure 5 and 6.

L179-180: “In spring, the relationships were positive between the visible and RE1 reflectance and elevation while negative for RE1-NIR2 (Fig. 3).”. In the figure 3 I can see “red”, RE1 and NIR1, but not NIR2. I suggest to check and adapt this part.

Figures 3 and 4: could you add the units for the axis? Moreover, please check the altitude values in the box “red-May 2nd” and “Green-October 14th”.

Figure 5 and 6: could you add the R2 in the figures to foster the comparison among them?

L231: could you use the acronym RE instead of red edge?

L391: could you replace “forest proprieties” with “forest structure”

Figure 17: the letters “a” and “b” in the figure are missed.

L482: Please use the acronym for red edge.

L486: Please use the acronym for red edge.

487: Please use the acronym for red edge.

L490: the site conditions considered in this study. Maybe the soil fertility or climate and microclimate variables could impact on the reflectance.

L493: Please use the acronym for red edge.

L499-515: this part is more appropriate for the discussion section.

L502: pure and even-aged stands, rights?

Reviewer #2: Review of the manuscript number PONE-D-20-38051: ‘Evaluating the effect of stand properties and site conditions on the forest reflectance from Sentinel-2 time series’ submitted for publication for PlosOne.

The manuscript investigates the effects of forest stand- and environmental characteristics for three selected tree species and their reflectance in individual Sentinel-2 bands. Species of interest are common beech, silver fir and Scots pine, and study stretches over 16 collection dates (3 in 2018& 13 in 2019).

The manuscript is well structured and well written, and seems correctly elaborated from the technical point of view. It includes most of necessary methodological details, although I suggest including more details and simplifying the Methods structure (below) to assure replicability of the study.

However, I missed the clear specification of the study novelty and application of the findings. Manuscript builds on publications from 1980’ to relate the reflectance and the tree species (Roy, 1989), which provides a great introduction into remote sensing of vegetation for novices. Yet, higher inclusions of the state-of-art publications would help to narrow down the current research gap, and specify how the study advances our understanding between the stands structural characteristics and reflectance. This relates to very limited overall applicability of the study findings beyond the fact that the patterns are diverse. I think that specification of novelty would greatly help to exemplify its advancements and new questions that needs to be answered.

Although the novelty specification is missing from the Introduction, authors brings some very interesting Results regarding effects of deciduous broadleaved/coniferous understory (L216, L431), effect of age (L232), crown closure (Fig. 9), link with phenology (as the main effect of elevation,…), etc. Therefore, I encourage authors to tackle better the novelty of their approaches, and how their findings complete and advance our knowledge in those particular questions. Nice overview is summarized in Conclusions (L502-515); yet, this belongs to Discussion. Instead, avoid in Discussion general statements and limit the Conclusions to overall study implication. Most importantly, avoid the general statements about usefulness of Sentinel-2 (L479-480). I agree that is it useful, but that is why it was launched 5 years ago (1), and it is clearly not this study novelty.

I list further notes regarding the visual representation of the study, investigation techniques, etc. Overall, I think the study has a great potential to be highly valuable and informative; but needs to be clearer in its novelty.

References:

Meyer, L.H., Heurich, M., Beudert, B., Premier, J., Pflugmacher, D., 2019. Comparison of Landsat-8 and Sentinel-2 data for estimation of leaf area index in temperate forests. Remote Sens. 11, 1–16. https://doi.org/10.3390/rs11101160

Roy, P.S., 1989. Spectral reflectance characteristics of vegetation and their use in estimating productive potential. Proc. Plant Sci. 99, 59–81. https://doi.org/10.1007/BF03053419

Major comments

Statistical evaluations

If I understand correctly, study analyses 7 independent variables (stand + environment) * 10 bands * 16 dates = did you evaluated 1120 gam models? Please, specify in Methods (‘e.g. we constructed gam model for each combination of XX & Y for each date..’). Models are further evaluated only by Adjusted R squared. Please specify why you have selected this criteria, and not e.g. AIC . For an example of evaluation and visualization of R-squared, please see e.g. (Meyer et al., 2019).

Plotting:

I suggest moveing Tab. 2 and Tab. 3, and Figs. 3,4,7,8,9,15 into Supplementary material and replace them with graphic summarizing R2 values, or selecting most relevant bands. Also I suggest to move the reflectance curves (Fig. 8&15) into supplementary material, and in the main text present only the most important NIR2-SWIR2 bands for different understories. I think that the reflectance curves are in general very useful, consider their potential to directly include them into some open-access repositories /Spectral Libraries (e.g. https://ecosis.org/, please check applicability).

Minor comments

Abstract

L 12-14 reformulate the sentence. I think it is a high ‘frequency’ allowing quantification of XXX (specify)

L 38 = cited work are from 2004-2005. how did your study advanced these knowledge? What is the current state-of art? complete the most recent advances on the topic. Specify

Introduction

L48 – can replace red edge by RE acronym?

L 59 – how does needle age relates to tree age? Does not tree changes needles every 3-4 years?

L94-95 – what is the role/purpose/potential of increased ‘frequency’ here? Do you is it to ‘tract down the development over the year/’vegetation season, etc.? Be specific. Is it that maybe this increased frequency can help to differentiate understory before beech foliation, for example?

Methods

L133-134 – what is the resolution of the ‘subareas’? Can ‘subareas’ be considered as forest stands, to keep understandable terminology?

L136: also included ‘tree species’? Complete. Or, simply add “in addition, the information used …” to make sure that readers understands that you have the species information

Please move the 2.4.1 part into 2.2. Section and accordingly 2.4.2 into Reference data subsection, and this information should be specified at the first mention to improve the structure and provide reader all necessary details within Data collection chapter. No need to have ‘2.4 Methods’ within ‘Methodology’, just call it ‘Data processing’.

L161-162 – ‘mean reflectance values were extracted to polygons’? Why this is needed? Did not you just added the mean reflectance by subarea (= polygon) by using ID to specific sub-areas site and stand conditions?

Section 2.4.3 = could be this section included simply in ‘Regression models’ section, as a data pre-processing/data-cleaning?

L170-174 – how have you grouped the data? Or, were the seasonality coded as a qualitative variable? Or, have you created a model for every date? (as seems from results?). Complete the software used, and package. Also, specify why later on just some of results are shown, and not for ach band, date, etc. Specify. Maybe add to Results that ‘only the statistically significant predictors are shown’ if this is the case?

L174: was it ‘adjusted R’?

Results

L216 – lower reflectance: in which band?

Discussion

L395 = higher reflectance in which band?

L413-414 – interesting finding; please clarify that you have known understory types and proportion of beach/oak from coniferous/broadleaved ratio, as missing until here and it is very interesting finding

Conclusion

keep conclusions short

Tab. 1 – Stand density description: do I get this right that unit ‘4’ represent that there is 4 times more volume then fully stocked stand? What is me meaning of the 0-10 for coniferous/broadleaves? 1 = fully coniferous, 10 = fully broadleaved? Complete.

Fig. 2- complete ‘.. in years 2018 and 2019’. Also, add in Methods that you combine the 2018 and 2019 dates to have higher scenes frequency within a year, while assuming that the reference stand data did not changed between 2008-2019, right?

Tab. 2 Caption: “model’ or ‘models’? Complete meaning of numbers in bolt, same for tab. 3

Fig. 3 – correct x label axis in Red plot, May 2nd?

Fig. 6 explain meaning of the dashed and solid lines?

Fig. 16 – what is the differences between a-c? Consider to remove the background to exemplify that the lines are moving if this is the case (as the unfolding elevation), or in some other way clarify the message of the fig? Can you link better the reflectance and phenology? this can be also one of the main traits of the manuscript.

GAM results plots: consider to replace the dashed lines by the shading, and adjust the y (smooth (x)) value to include the intercept to improve the interpretability of the partial effect, i.e. in R library mgcv(plot(gam_model, shift = coef(gam_model)[1]))

why the Tab 2 and 3 have different number of dates, the same for the Fig3&4 and Fig.8? Why you have mentioned all dates (Fig.2) and later just few dates are used?

Reviewer #3: The study analysed the influence of different forest stand parameters and site conditions on forest stand reflectance. The effect of stands and site properties on reflectance in different parts of the growing season was captured using the dense time series provided by Sentinel-2 imagery. It must be appreciated that the authors conducted their research in mountainous area of Carpathians. There is a lack of this kind of studies in a such complicated terrain. Especially valuable are findings on the time of the year at which forest characteristics can be easier to distinguish.

Structure of the article is good, text is clear. All necessary data are available. Discussion and results are clearly stated.

However, more details should be provided on these issues:

- what exactly is meant by pre-processing of Sentinel 2 images using sen2r (page 9, line 151)? Is it what is described in related paragraph or something else? Text here is not clear.

- some descriptive statistics on selected forest stands should be provided: what is the size of an average "subarea"? Few hectares? What is "sample"? A pixel? It is not clear. (page 9, chapter 2.4.2)

- it would be probably more appropriate to conclude that the reflectance is influenced by actual phenophase instead "reflectance was mainly influenced by the elevation" (page 29, line 484). Despite the fact the altitudinal distribution of phenophases during spring and autumn is controlled by elevation, it is phenophase, not the elevation itself, what determines the reflectance

6. PLOS authors have the option to publish the peer review history of their article (what does this mean?). If published, this will include your full peer review and any attached files.

Reviewer #1: **Yes: **Giovanni Santopuoli

Reviewer #2: **Yes: **Mária Potterf

Reviewer #3: No

---

## [Author Response · Author response to Decision Letter 0]

13 Jan 2021

Reviewer #1

Dear Giovanni Santopuoli, 

Thank you for your valuable comments and suggestions. We revised the manuscript according to your and other reviewers’ comments. We hope the corrections of the manuscript meet your expectations.

The main changes include:

• We rearranged the methods part and provided more precise description of particular steps;

• We rearranged the results section and removed/joined part of the figures to increase its clarity and readability;

• We shortened the conclusions part.

Specific comments are addressed below:

• However, I think that 17 figures are too much for a scientific publication. I suggest to report some data in the table format, particularly those data about the comparison among variables (e.g. aspect, slope and elevation) for each tree species. 

Thank you for your comment. We agree that the 17 figures were too much. Therefore, we l, we joined figures 3 and 4 (now figure 3); 5 and 6 (now figure 5), 13 and 14 (now figure 10). We also removed figures 9 and 11 as they did not provide much new information.

However, the plots comparing the influence of particular variables (GAM partial plots) are, in our opinion, very valuable as they show the relationships between predictors and allow to compare them. We added additional information into figures description to increase its interpretability: 

GAM results – visualization of partial effects on reflectance (…). The solid line represents the relationship between predictor and response variables, and the light blue shaded area represents confidence intervals. 

Additionally, we created one table (Table 3) with the selected values of adjusted R2 for all three examined species and moved the two original tables into supplementary material. 

In current form, the manuscript includes 12 figures. 

• The introduction is well written, while material and method could be improved. In particular, the description of study area could be improved and the description of the selected images, and the pre-processing details could be enriched.

Thank you! We provided a more detailed description of the study area (lines 133-147), analyzed stands (lines 174-176), selected images (154-156), and methodology of pre-processing (lines 162-163).

• The result section is a bit confused, due to the abundance of figures. I suggest to reorganize the text to follow a linear description starting with the comparison among variables and then focusing on the variables that resulted more impacting on the tree species reflectance.

Thank you for your comment, we removed part of the figures and joined some of them (as described above) to increase the readability and clarity of the results section. 

We rearranged some parts of the results; however, the goal was to analyze the effect of single variables and then develop a GAM model to consider all significant variables' impact at once. We hope it’s now more clear and easier to follow. 

• The discussion is fine, even if I suggest to enrich the section with the text reported in the conclusion (Lines 499-515), because the conclusion, in my opinion, should be more concise and focused on the findings obtained.

Thank you for your comment, we moved the part that you mentioned into the discussion part. Also, we shortened the conclusion part (it is now in lines 515-533).

• Finally, please check for the use of acronyms in the whole text. 

OK, corrected.

• L39: “openness of trees” do you mean “tree density” or “canopy gaps/closure”? 

By “openness of trees” we meant “canopy closure”. We changed that in the text (line 41).

• L61: please use the acronym since you already introduced it in the line 47.

Thank you, it was corrected.

• L55-74: very nice introduction. However, I suggest to add few words for the uneven-aged stands. In this text you focused on the even-aged stands describing how the reflectance differs between young and old trees, and between broadleaved and conifer species. Is it possible to discriminate the reflectance of uneven-aged and mixed forest stands through remote sensing data?

Thank you for your comment. In our study, we analyzed pure and, when age was one of the predictors, even-aged forest stands. Moreover, our paper is already quite extensive and focused on many different aspects. Therefore we didn’t include that topic in the introduction; however, this problem is mentioned in the discussion part in lines 497-503, where we also highlighted that uneven-aged stands should be analyzed in further studies.

The possibility of using Sentinel-2 in discriminating uneven-aged and mixed species composition would depend largely on the forest type and structure (e.g. if the younger age classes are a part of an understory or lower story), what/how many tree species occurring and how they are grouped/mixed etc. However, as we mentioned before, it is the topic for another study. 

• L118-120: please use the same style to introduce the tree species and for references. 

The style was unified.

• L114-120: how did you select the 13000 km2? Is it included in an administrative region or a different administrative border? It is not clear which is the criterion used to select the study area. If possible, please add more information about forest ownership, forest management system, stand age and other information useful to describe the forest stands. 

Thank you for your comment, we selected the study area based on Forest District units (which is a basic economic/organizational unit in the structure of Polish State Forests). These forests districts (in total 28) were selected based on two criteria: 1) laying in the Carpathians/Carpathians foothills/Outer Subcarpathia region and 2) laying within the area of four examined Sentinel-2 tiles. 

We added that information in lines 133-137.

In our opinion, this area is representative and large enough to draw conclusions on the site conditions/stand properties influence on forest stand reflectance. 

Regarding the characteristics of analyzed forests, the study was conducted based on the Polish State Forests database. Therefore, only public forests were analyzed (it is mentioned in line 169). We also provided information about examined forest stands age in lines 174-176. 

Forest management systems in the research area are adjusted to the forest species composition. In the beech-dominated stands, sherlterwood system is used, silver fir dominated stands are managed by stepwise cutting, whereas Scots pine stands are mostly managed with a clear-cutting system. This information was added in lines 143-147.

• L123: Could you explain why only 3 images were downloaded for the 2018? 

We only selected 3 images from 2018 as the spring (May) imagery was missing in 2019, mainly due to high cloud coverage during this time. We assumed, that information from these part of the growing season will be very valuable, particularly when analyzing environmental conditions impact on beech stands reflectance. We added that information in lines 154-156.

• L128-129: this is the second time that you introduced the acronyms RE, NIR, and SWIR. Please check in the instructions for the authors about the use of acronyms. 

Ok, corrected.

• L155: can you better explain the masking procedures (i.e. the R package used). Is it again sen2r? 

We applied our own cloud-masking algorithm in R – based on the land cover classification provided with Sentinel-2 product. We mask (cut off) areas marked as clouds, clouds shadows or snow using a raster package. We added that information in line 163.

• L158: “12,235 samples were selected” from where? Are they randomly generated by the authors? Are they national forest inventory samples? Could you describe better how you select the samples? 

We used data from Forest Data Bank – out of all subareas (stands) for the study area, only stands with a 100% share of a particular tree species were selected – i.e. 12,235 subareas. It might be confusing that we used the word “samples” here; therefore, we changed it to “stands”. It is described now in lines 171-174.

• L177: you stated that “elevation was the most important driver for beech stand…”. Is it possible to show the differences respect to other site conditions (i.e. slope, aspect)? Otherwise I can’t see this statement. Maybe you could use here the text of lines 200-206 and figure 5 and 6.

Thank you for your comment, we added the reference to Table 2 and Figures 3 and 5. We hope it’s now better justified that elevation is indeed the most important driver. 

• L179-180: “In spring, the relationships were positive between the visible and RE1 reflectance and elevation while negative for RE1-NIR2 (Fig. 3).”. In the figure 3 I can see “red”, RE1 and NIR1, but not NIR2. I suggest to check and adapt this part.

Bands from RE2-NIR2 region (i.e. RE2, RE3, NIR1, NIR2) are characterized by a very similar spectral behavior and patterns; therefore, we selected one band from this range to show the relationships. We added that information in lines 212-213. 

• Figures 3 and 4: could you add the units for the axis? Moreover, please check the altitude values in the box “red-May 2nd” and “Green-October 14th”. 

Thank you for your comment; we corrected plots. The elevation units are provided now in figure description (now it is joined in Figure 3). 

• Figure 5 and 6: could you add the R2 in the figures to foster the comparison among them? 

R2 was added to the figure titles. Now it’s figure 5, and R2 was also added to figures 7 and 10.

• L231: could you use the acronym RE instead of red edge? 

Yes, we corrected that in the whole text. 

• L391: could you replace “forest proprieties” with “forest structure” 

Corrected.

• Figure 17: the letters “a” and “b” in the figure are missed. 

Corrected.

• L482: Please use the acronym for red edge.

• L486: Please use the acronym for red edge.

• 487: Please use the acronym for red edge.

Corrected.

• L490: the site conditions considered in this study. Maybe the soil fertility or climate and microclimate variables could impact on the reflectance. 

Thank you, we added that to the sentence (line 527).

Regarding the climate/microclimate variable, a microclimate is strongly related to topography, which characterize climate conditions (Monserud and Rehfeldt, 1990). As the indirect measures of regional climate variations, we used elevation above sea level. Local microclimate variation was also characterized with aspect and slope. Because the site productivity in the analyzed area is mainly related to the elevation above sea level, the analyzes did not include soil fertility.

We added that information in the methodology section in lines 179-182.

• L493: Please use the acronym for red edge. 

Corrected.

• L499-515: this part is more appropriate for the discussion section. 

Thank you for your comment, we moved that part into discussion section.

• L502: pure and even-aged stands, rights? 

In case of analyzing the age influence on reflectance, indeed, only even-aged stands were considered (we added that information – lines 497-498).

Reviewer #2

Dear Mária Potterf,

Thank you for your valuable comments. According to your suggestions, as well as other reviewers comments, we revised the manuscript. 

The main changes include:

• We provided more detailed study novelty and applicability description in the Introduction part;

• We rearranged the methods part and modified the description of particular steps (especially, how the models were calculated/selected);

• We rearranged the results section and removed or joined part of the figures to increase its clarity and readability;

• We shortened the conclusions part.

Specific comments are addressed below:

• However, I missed the clear specification of the study novelty and application of the findings. Manuscript builds on publications from 1980’ to relate the reflectance and the tree species (Roy, 1989), which provides a great introduction into remote sensing of vegetation for novices. Yet, higher inclusions of the state-of-art publications would help to narrow down the current research gap, and specify how the study advances our understanding between the stands structural characteristics and reflectance. 

Thank you for your comment. The novelty of our study was indeed insufficiently provided in the introduction part. 

Although the general principles of reflectance variability of forests stands are well known, the detailed information on how they change in space and time, depending on different environmental/stand properties for larger areas is missing. All these drivers influencing stand reflectance are usually analyzed separately, while here we provided multivariate analysis approach. On the other hand, in studies focused on predicting some properties based on reflectance, also usually one variable/variables group is taken into consideration. However, in our opinion, the description of these general principles of the reflectance variability, which were in many cases described in older publications, should still be included in the introduction part. 

To our knowledge, there are no studies taken into consideration both stand and site properties variables. Furthermore, as you mentioned, the results of our analysis provided new, interesting information on, for example, understory vegetation impact on SWIR. We emphasized the potential of SWIR, as well as RE bands in the introduction part. We also included some new publications (for example (Chrysafis et al. 2017; Astola et al. 2019; Markiet and Mõttus 2020). However, there are not many new studies analyzing how specific site or stand property impacts the stand reflectance in different wavelengths. On the contrary, there are many studies on predicting/mapping different forest structure properties – for example (Astola et al. 2019). These studies are also often based on indices derived from bands, not bands themselves. Therefore, in many cases, it’s hard to obtain information from these newer studies that could be useful from the point of view of our research – i.e. specific information how variable influence the analyzed reflectance region.

The description of the need for our study and the findings application is now provided in the paragraph in lines 102-118. And the novelty is also highlighted in lines 123-125.

• This relates to very limited overall applicability of the study findings beyond the fact that the patterns are diverse. I think that specification of novelty would greatly help to exemplify its advancements and new questions that needs to be answered.

Thank you for your comment, we complete the Introduction section and provided specification of applicability and novelty, as described in the answer above.

Regarding the applicability of the study findings, understanding the impact of forest stand properties and site conditions on the forest reflectance is crucial in applying remote sensing technologies in forests monitoring. Linking the diversity of forest reflectance caused by different stands and site characteristics enables applying the satellite remote sensing in forest inventory. Thanks to this, satellite imagery can be used to determine the features of stands, such as species composition, density, growing stock volume, biomass and age. The research results may also be of importance in the assessment of forest sites, particularly when analyzing large areas with various conditions. This is described in lines 106-118.

• Although the novelty specification is missing from the Introduction, authors brings some very interesting Results regarding effects of deciduous broadleaved/coniferous understory (L216, L431), effect of age (L232), crown closure (Fig. 9), link with phenology (as the main effect of elevation,…), etc. Therefore, I encourage authors to tackle better the novelty of their approaches, and how their findings complete and advance our knowledge in those particular questions. 

The research novelty consists of the simultaneous consideration of the impact of site conditions, the structure of stands, the age of trees and understory vegetation on both deciduous and coniferous forests reflectance.

In the introduction, the research novelty was mentioned (lines 123-125).

Applicability of study findings is described in lines 106-118.

• Nice overview is summarized in Conclusions (L502-515); yet, this belongs to Discussion. Instead, avoid in Discussion general statements and limit the Conclusions to overall study implication.

Thank you for your comment, we modified and shortened the conclusions chapter, and the part that you mentioned was moved to the discussion section. 

• Most importantly, avoid the general statements about usefulness of Sentinel-2 (L479-480). I agree that is it useful, but that is why it was launched 5 years ago (1), and it is clearly not this study novelty.

Thank you for your comment. We removed that part and modified the conclusion part (now lines 515-533). 

• Statistical evaluations If I understand correctly, study analyses 7 independent variables (stand + environment) * 10 bands * 16 dates = did you evaluated 1120 gam models? Please, specify in Methods (‘e.g. we constructed gam model for each combination of XX & Y for each date..’). Models are further evaluated only by Adjusted R squared. Please specify why you have selected this criteria, and not e.g. AIC . For an example of evaluation and visualization of R-squared, please see e.g. (Meyer et al., 2019). 

Thank you for your comment. We agree that it should be described in more detail. 

We did not evaluate all 1120 GAM models - firstly, in the preliminary analysis, the most important variables and dates were considered (it is now noted in subchapter 2.4. Regression models). Also, as GAM models can handle multiple predictor variables in some cases, multiple regression was performed in our study (we added that information in line 202).

The total number of fitted models is hard to specify, as part of them were constructed only for the most important variables/dates; however, we specified which models were built for each species in Table 2.

Also, we provided a new table with only the most important relationships and corresponding adjusted R2 values in the results section (Table 3) from previous tables 2 (beech) and 3 (fir), with information on pines relationships R2 also. 

Regarding the model evaluation, after (Gwowen 2008) we assumed that adjusted R2 is appropriate when evaluating model fit and comparing alternative models in the feature selection stage of model building. In remote sensing studies, the variation explained by only the independent variables that affect the dependent variable is frequently described by adjusted R2. AIC is a criterion for selecting between statistical models with different numbers of predictors, which is used because a model with more predictors generally produces more accurate predictions, but is also more likely to overfitting. A more accurate method of checking for overfitting a model, however, is to use cross-validation. The main research goal was not to build predictive models but to identify variables that modify the forest reflection. Therefore, the adjusted R2 was used to select the most significant variables. 

• Plotting: I suggest moveing Tab. 2 and Tab. 3, and Figs. 3,4,7,8,9,15 into Supplementary material and replace them with graphic summarizing R2 values, or selecting most relevant bands. Also I suggest to move the reflectance curves (Fig. 8&15) into supplementary material, and in the main text present only the most important NIR2-SWIR2 bands for different understories. I think that the reflectance curves are in general very useful, consider their potential to directly include them into some open-access repositories /Spectral Libraries (e.g. https://ecosis.org/, please check applicability). 

Thank you for your comment, we moved part of the tables (2,3) and to the supplementary materials. We provided one table with the most relevant bands/dates/predictor variable and corresponding R2 values (it is now Table 3).

Regarding figures, we joined figures 3 and 4 (now figure 3); 5 and 6 (now figure 5), 13 and 14 (now figure 10); We also removed figures 9 and 11 as they did not provide much new information.

However, in our opinion, the reflectance curves are very informative, so we decided to leave them in the main text. 

Thank you for your suggestion to provide the spectral signatures into open-access repositories, we will consider that.

• L 12-14 reformulate the sentence. I think it is a high ‘frequency’ allowing quantification of XXX (specify) 

Thank you, we removed that part of the sentence. 

• L 38 = cited work are from 2004-2005. how did your study advanced these knowledge? What is the current state-of art? complete the most recent advances on the topic. Specify 

Thank you for your comment, in the paragraph you mentioned we wanted to introduce general principles of vegetation/stand reflectance; therefore, some of the cited studies are older.

We also included some new publications according to your suggestion (for example (Chrysafis et al. 2017; Astola et al. 2019; Markiet and Mõttus 2020). However, as we mentioned in answer to your comment above, there are not many new studies analyzing how the specific site and stand properties impact the stand reflectance in different wavelengths. On the contrary, there are many studies on predicting/mapping different forest structure properties – for example (Astola et al. 2019). These studies are often based on indices derived from bands, not bands themselves. Therefore, in many cases, it’s hard to obtain information from these studies that could be useful from the point of view of our study – i.e. specific information how variable influence the analyzed reflectance region.

• L48 – can replace red edge by RE acronym? 

We introduced RE acronym, which was subsequently used through the text.

• L 59 – how does needle age relates to tree age? Does not tree changes needles every 3-4 years? 

In the case of conifer trees, the leaf area index (LAI) is correlated with the age of trees - with older stands, LAI increase, see (Niemann 1995) and (Nilson and Peterson 1994).

Regarding needle age impact on canopy reflectance, it’s a complex task –mature needles, i.e. > 1 y.o. represent the majority, however e.g. the new needles are mainly distributed at the upper part of crowns (see Wu et al. 2018).

• L94-95 – what is the role/purpose/potential of increased ‘frequency’ here? Do you is it to ‘tract down the development over the year/’vegetation season, etc.? Be specific. Is it that maybe this increased frequency can help to differentiate understory before beech foliation, for example?

The increased frequency, i.e. higher temporal accuracy is important to examine relationship during different parts of the season, as you mentioned, for example – the understory can develop just within few days; therefore imagery acquired within a short period may differ considerably. The role of increased temporal accuracy can be particularly observed at the end of April/beginning of May when the changes are very rapid. We added that information in lines 91-93. Furthermore, with the increased temporal resolution, there is a higher chance for obtaining cloud-free imagery. 

• L133-134 – what is the resolution of the ‘subareas’? Can ‘subareas’ be considered as forest stands, to keep understandable terminology? 

As subarea is a forest fragment uniform in terms of economic features (e.g. age, species composition), then yes it can be treated as a forest stand. We changed the term “subarea” to “stand”.

We also provided information about mean stand sizes (line 174).

• L136: also included ‘tree species’? Complete. Or, simply add “in addition, the information used …” to make sure that readers understands that you have the species information

Yes, subareas (i.e. stands) also contain information on tree species, we added that in line 171.

• Please move the 2.4.1 part into 2.2. Section and accordingly 2.4.2 into Reference data subsection, and this information should be specified at the first mention to improve the structure and provide reader all necessary details within Data collection chapter. No need to have ‘2.4 Methods’ within ‘Methodology’, just call it ‘Data processing’. 

Thank you for your comment, we moved these parts according to your suggestions. Now the structure of methods chapter is as follows: 

2.1 Study area

2.2 Satellite imagery collection and pre-processing

2.3 Reference data

2.4 Regression models

• L161-162 – ‘mean reflectance values were extracted to polygons’? Why this is needed? Did not you just added the mean reflectance by subarea (= polygon) by using ID to specific sub-areas site and stand conditions? 

The statistical unit in the analyzes was the single stand; therefore, the mean reflectance values were assigned to each polygon representing the stand. Mean reflectance had to be extracted, as stands polygon represent more than one Sentinel-2 pixel.

• Section 2.4.3 = could be this section included simply in ‘Regression models’ section, as a data pre-processing/data-cleaning? 

Thank you for your comment, as mentioned above, we rearrange Chapter 2, and subchapter 2.4 now consist of preliminary analysis for regression models. 

• L170-174 – how have you grouped the data? Or, were the seasonality coded as a qualitative variable? Or, have you created a model for every date? (as seems from results?). Complete the software used, and package. Also, specify why later on just some of results are shown, and not for ach band, date, etc. Specify. Maybe add to Results that ‘only the statistically significant predictors are shown’ if this is the case? 

Thank you for your comment, we created models separately for different dates, so seasonality was not coded as a qualitative variable. We provided new Table 2 with GAM models that was fitted. We provided software used (mgcv package; line 203). 

Only selected results are shown, as most of the models were statistically significant, but we wanted to present variables in bands/dates that explain the largest part of reflectance variability. At the same time, we had to reduce the results in some sections, as for example, bands from the RE2 to NIR2 range are characterized by very similar spectral patterns, (we mentioned that in lines 212-213), therefore in case of beech stands only selected bands are shown in Fig. 3. Similarly in case of silver fir stands, the general patterns of relationships between age and reflectance are similar throughout the year (lines 264-265), so one date is presented in Fig. 6. 

• L174: was it ‘adjusted R’? 

Yes, we added that information (line 201).

• L216 – lower reflectance: in which band? 

We meant lower reflectance in general, added (line 234).

• L395 = higher reflectance in which band? 

It is specified in the same sentence (“the entire growing season and in all wavelengths”). Now it is line 414. 

• L413-414 – interesting finding; please clarify that you have known understory types and proportion of beach/oak from coniferous/broadleaved ratio, as missing until here and it is very interesting finding 

It is described now in the 2.3 Reference data in Table 1.

• keep conclusions short 

Thank you for your comment; we shortened the conclusions chapter. 

• Tab. 1 – Stand density description: do I get this right that unit ‘4’ represent that there is 4 times more volume then fully stocked stand? What is me meaning of the 0-10 for coniferous/broadleaves? 1 = fully coniferous, 10 = fully broadleaved? Complete. 

Regarding the stand density – yes you’re right, but as these cases are extremely rare and it might be confusing, we removed that from table 1 and just added the information rarely above 1. 

We explained the meaning of understory share in Table 1. 

• Fig. 2- complete ‘.. in years 2018 and 2019’. 

OK, it was corrected.

• Also, add in Methods that you combine the 2018 and 2019 dates to have higher scenes frequency within a year, while assuming that the reference stand data did not changed between 2008-2019, right? 

Yes, we assumed that the examined stands did not change from 2018 to 2019. However, if the change did occur (i.e. cutting), the spectral values would be removed as outliers.

• Tab. 2 Caption: “model’ or ‘models’? Complete meaning of numbers in bolt, same for tab. 3

Thank you for your comment, we moved these two tables to the supplementary material and specified this information. 

• Fig. 3 – correct x label axis in Red plot, May 2nd? 

Corrected. 

• Fig. 6 explain meaning of the dashed and solid lines? 

We added that information in GAMs figures description:

GAM results – visualization of partial effects on reflectance (…). The Solid line represents the relationship between predictor and response variables and the light blue shaded area represents confidence intervals. 

• Fig. 16 – what is the differences between a-c? Consider to remove the background to exemplify that the lines are moving if this is the case (as the unfolding elevation), or in some other way clarify the message of the fig? Can you link better the reflectance and phenology? this can be also one of the main traits of the manuscript.

The differences between a-c are in the elevation of leaf unfolding (in the background, you can see the Sentinel-2 image in true-color composition); the color lines represent contour lines (isohypses). We modified the figure description (now it’s Figure 11).

• GAM results plots: consider to replace the dashed lines by the shading, and adjust the y (smooth (x)) value to include the intercept to improve the interpretability of the partial effect, i.e. in R library mgcv(plot(gam_model, shift = coef(gam_model)[1])) 

Thank you for your comment, we replaced the dashed lines by the shading and modify the y axis - it has now the same range to improve the interpretability. 

• why the Tab 2 and 3 have different number of dates, the same for the Fig3&4 and Fig.8? Why you have mentioned all dates (Fig.2) and later just few dates are used? 

Thank you for your comment, as explained above, we did not use all of the dates for the further, GAM model fitting (it is now described in Table 2) and in lines 197-203 in the methodology section. For example, in the case of beech stands, the influence of elevation was not observed during summer; therefore, there was no point taking into consideration the summer dates. 

The results were further reduced - only selected bands and dates are presented, as most of the models were statistically significant, but we were able to present most important variables in bands/dates that explain the largest part of reflectance variability. At the same time, we had to reduce the results in some parts, as for example, bands from the RE2 to NIR2 range are characterized by very similar spectral patterns (see the example of beech in lines 212-213). Therefore, the figures 3 and 4 which were joined (it is now figure 3) present only selected bands. Still, in our opinion it is the very informative figure – it shows that the changes in beech reflectance vary both depending on band and time of the year and, furthermore, these changes can occur within just a few days. On the other hand, in case of fir, figure 8 shows the slight reflectance changes with age in different bands, but these patterns are very similar throughout the year; therefore only one date was selected and shown.

Reviewer #3

Dear Reviewer,

Thank you for your valuable comments and suggestions. We revised the manuscript according to your and other reviewers’ comments. 

The main changes include:

• We rearranged the methods part and provided more precise description of particular steps;

• We rearranged the results section and removed/joined part of the figures to increase its clarity and readability;

• We shortened the conclusions part.

Specific comments are addressed below:

• what exactly is meant by pre-processing of Sentinel 2 images using sen2r (page 9, line 151)? Is it what is described in related paragraph or something else? Text here is not clear. 

We rephrased and moved this paragraph (please see lines 153-163).

• some descriptive statistics on selected forest stands should be provided: what is the size of an average "subarea"? Few hectares? What is "sample"? A pixel? It is not clear. (page 9, chapter 2.4.2)

Thank you for your comment, we provided more detailed information about selected stands in lines 174-176. Furthermore, we changed the word “sample”, which was confusing, to “stands” (as we simply refer to stands here as well).

• it would be probably more appropriate to conclude that the reflectance is influenced by actual phenophase instead "reflectance was mainly influenced by the elevation" (page 29, line 484). Despite the fact the altitudinal distribution of phenophases during spring and autumn is controlled by elevation, it is phenophase, not the elevation itself, what determines the reflectance 

We rephrased the sentence to (lines 519-520) :

The main driver of beech stands reflectance variability was the elevation, which strongly affects the phenophases (...).

---

## [Decision Letter · Decision Letter 1]

9 Feb 2021

PONE-D-20-38051R1

Evaluating the effect of stand properties and site conditions on the forest reflectance from Sentinel-2 time series

PLOS ONE

Dear Dr. Grabska,

Thank you for submitting your manuscript to PLOS ONE. After careful consideration, we feel that it has merit but does not fully meet PLOS ONE’s publication criteria as it currently stands. Therefore, we invite you to submit a revised version of the manuscript that addresses the points raised during the review process.

The reviewers are now mostly satisfied with your revision, but one them still requires minor changes to consider before accepting for publication. Please, read the reviewer's suggestions and address them as much as possible. 

We look forward to receiving your revised manuscript.

Kind regards,

Michal Bosela, Ph.D.

Academic Editor

PLOS ONE

Reviewers' comments:

Reviewer's Responses to Questions

**Comments to the Author**

1. If the authors have adequately addressed your comments raised in a previous round of review and you feel that this manuscript is now acceptable for publication, you may indicate that here to bypass the “Comments to the Author” section, enter your conflict of interest statement in the “Confidential to Editor” section, and submit your "Accept" recommendation.

Reviewer #2: All comments have been addressed

2. Is the manuscript technically sound, and do the data support the conclusions?

Reviewer #2: Yes

3. Has the statistical analysis been performed appropriately and rigorously? 

Reviewer #2: Yes

4. Have the authors made all data underlying the findings in their manuscript fully available?

Reviewer #2: No

5. Is the manuscript presented in an intelligible fashion and written in standard English?

Reviewer #2: Yes

6. Review Comments to the Author

Reviewer #2: Review of the Manuscript PONE-D-20-38051R1 :Evaluating the effect of stand properties and site conditions on the forest reflectance from Sentinel-2 time series PLOS ONE

The manuscript now nicely and clearly specifies the advances of the current research, and how it addresses ongoing research gaps (L87-89, 102-105, etc.).Methods section was completed and now assures reproducibility. Well done!

For Results sections, I list some more suggestions about how further improve data visualizations. I suggest removing multiple plots, as they do not show a clear message and therefore are difficult to interpret. E.g. from a reader perspective, I do not see why only selected ‘dates’ were plotted, and not the others (was that because of highest R squared/random selection/ something else)? As those Figs. do not show very clear message besides that they are different, I suggest removing them to keep the manuscript concise. Please see details for further information, and consider implementing them in the manuscript.

Abstract

L18-19 – I suggest formulation: “Our study aim(ed) to quantify the site and forest parameters affecting …” instead of this rather ‘conclusive’ statement, as at the beginning of Abstract you show reader what your study aimed to explore

L72-74 , 87=89– well done! nice, clear and specific summary of previous sentences. Good example of a good writing :)

Methods:

L197-204 – do I get this right that dependent variables are all individual 11 bands of Sentinel-2? Please specify.

Results

L217 – what is meant here by the ‘changes in reflectance’? were you predicted an absolute change in e.g. SWIR values between two dates (i.e. May 2 to May 12)? I think no as one glm was fit for each band and therefore here you report the change in adjusted R squared? Please be specific.

Discussion

L474 = Fig. 9 remove bold fond

Table 3: I like this specifications, it is clear. However, you might consider to use something like ‘correlation heat maps’ to show correlation for individual bands/over time if you wish to visualize changes in R2? e.g. http://www.sthda.com/english/wiki/ggplot2-quick-correlation-matrix-heatmap-r-software-and-data-visualization This is just a suggestion for the future reference. This could replace Fig. 3 which I suggest to remove or move to Supp. material.

Tab. 1 = Later in the text it reads that understory can contain fir as well. So, does 1 = 100% of fir trees. Are oak and beech considered therefore in one class? (broadleaved?). Please clarify.

Fig.4: Use the same y limits to allow visual comparison between different dates. However, I think this image is very hard to interpret, and keep in mind how curves are shifting over time. I suggest to remove it or consider alternative ways of display. From my interpretation, all % understory follows the same pattern, but there is difference mostly in SWIR1 values. Consider choosing only extreme 0&50&100% understorey values to show SWIR1 values over time? x = time, y = reflectance, group = broadleaved understory. Would that tell your story better?

Fig. 6: also, I consider this unnecessary: not clear what is the main message of the plot, ale not clear why only NIR2 band values are shown and only on one date? The story would be very different if y-axis limits were the same. I like the interpretation thought: there is difference in reflectance for stands 0-50 and >50 but this could be likely visualized simply by boxplots for groups and tested. I suggest to remove Fig.6.

Fig. 8. Again, this plot is very difficult to interpret and see differences there. Crown closure should be a discrete variable (Table 1); here is instead plot as continuous. How would the story look like for the plot would be faceted by groups (~ crown closure)? Again, another dates are missing and it is unclear why only July 1st was selected?

Fig. 9 = also Fir was considered as understory dominant species?? Missing from Tab .1 ? Unclear. Plot 9 is not necessary.

Fig, 12 Very difficult to interpret. Again, there are 4 crown closure groups, but only one smooth line, difficult to spot the differences. I suggest to remove.

7. PLOS authors have the option to publish the peer review history of their article (what does this mean?). If published, this will include your full peer review and any attached files.

Reviewer #2: **Yes: **Maria Potterf

---

## [Author Response · Author response to Decision Letter 1]

15 Feb 2021

Dear Mária Potterf,

Thank you again for your valuable comments. According to your suggestions, we revised the results section. Please see the specific comments addressed below.

• The manuscript now nicely and clearly specifies the advances of the current research, and how it addresses ongoing research gaps (L87-89, 102-105, etc.).Methods section was completed and now assures reproducibility. Well done!

For Results sections, I list some more suggestions about how further improve data visualizations. I suggest removing multiple plots, as they do not show a clear message and therefore are difficult to interpret. E.g. from a reader perspective, I do not see why only selected ‘dates’ were plotted, and not the others (was that because of highest R squared/random selection/ something else)? As those Figs. do not show very clear message besides that they are different, I suggest removing them to keep the manuscript concise. Please see details for further information, and consider implementing them in the manuscript. 

Thank you for your kind comments. According to your suggestions we further improved the results section and removed or rearranged some figures. There are now 10 figures in the main text. We hope that now the results section is clearer and the remaining plots are easier to interpret. Please see the detailed responses below. 

• L18-19 – I suggest formulation: “Our study aim(ed) to quantify the site and forest parameters affecting …” instead of this rather ‘conclusive’ statement, as at the beginning of Abstract you show reader what your study aimed to explore 

Ok, it was corrected.

• L72-74 , 87=89– well done! nice, clear and specific summary of previous sentences. Good example of a good writing :) 

Thank you! 

• L197-204 – do I get this right that dependent variables are all individual 11 bands of Sentinel-2? Please specify.

Yes, the dependent variables are values in individual Sentinel-2 bands. 10 bands were used – as they are used in land applications (the ones with -10 and -20 meter resolution – it is described in lines 158-159). We specified information about dependent variables in lines 198-199.

• L217 – what is meant here by the ‘changes in reflectance’? were you predicted an absolute change in e.g. SWIR values between two dates (i.e. May 2 to May 12)? I think no as one glm was fit for each band and therefore here you report the change in adjusted R squared? Please be specific.

This description refers to what is, for example, shown in figure 3. The plots show the changes in the reflectance depending on the elevation above sea level for selected bands on different dates. For example, for Red there is a large difference between the average values of reflectance on given elevation between May 2, 7 and 12 (Fig. 3). We understood changes in reflectance as the changes in the observed and model values (red lines) between successive dates visible on subsequent figures in the row.

We also added to this sentence (line 218): the changes in reflectance depending on elevation, were very dynamic (...)

• L474 = Fig. 9 remove bold fond 

OK, corrected. 

• Table 3: I like this specifications, it is clear. However, you might consider to use something like ‘correlation heat maps’ to show correlation for individual bands/over time if you wish to visualize changes in R2? e.g. http://www.sthda.com/english/wiki/ggplot2-quick-correlation-matrix-heatmap-r-software-and-data-visualization This is just a suggestion for the future reference. This could replace Fig. 3 which I suggest to remove or move to Supp. material 

Thank you for your suggestion. Indeed, the correlation heat maps are a great way to visualize R2 or correlation coefficients. However, it would be difficult to produce one clear correlation plot in our case, as we have different bands as dependent variables, in different dates and for three different species. Therefore, in our opinion Table is more appropriate in this case. 

In our opinion scatterplots in Fig. 3 show the dynamic differences of broadleaved trees at the beginning and the end of the growing season, and therefore should stay in the main text. However, if you or editor insist, we can move it into the supporting information. 

• Tab. 1 = Later in the text it reads that understory can contain fir as well. So, does 1 = 100% of fir trees. Are oak and beech considered therefore in one class? (broadleaved?). Please clarify.

Thank you for your comment. In general, in the GAM modelling only broadleaved/conifer understory share was taken as the predictor variable. Therefore in the Table 1 we only have left information about coniferous/broadleaved understory. 

In this specific case of Scots pine, the understory species was diversified so it was possible to depict the differences in the spectral signatures. So additionally, for the purpose of visualization, we also used dominant species in case of Scots Pine stands understory. 

We added that information in the Table 1. 

• Fig.4: Use the same y limits to allow visual comparison between different dates. However, I think this image is very hard to interpret, and keep in mind how curves are shifting over time. I suggest to remove it or consider alternative ways of display. From my interpretation, all % understory follows the same pattern, but there is difference mostly in SWIR1 values. Consider choosing only extreme 0&50&100% understorey values to show SWIR1 values over time? x = time, y = reflectance, group = broadleaved understory. Would that tell your story better? 

Thank you for your comment. According to your suggestion, to improve interpretability, we produced new figure. It shows the reflectance variations depending on the broadleaved understory share only in SWIR1 band in selected dates during the season. However, the figure still consists the 10 different levels of broadleaved understory, but, in our opinion is now easier to visually compare. 

• Fig. 6: also, I consider this unnecessary: not clear what is the main message of the plot, ale not clear why only NIR2 band values are shown and only on one date? The story would be very different if y-axis limits were the same. I like the interpretation thought: there is difference in reflectance for stands 0-50 and >50 but this could be likely visualized simply by boxplots for groups and tested. I suggest to remove Fig.6. 

Thank you for your comment. Indeed, the figure was unclear and hard to interpret, therefore we replaced it with the new one. In new figure, the mean values for the 10-year age classes is presented in one plot. It clearly shows that the stands up to 50 years old are entirely separable in the region from RE2 to NIR2. 

• Fig. 8. Again, this plot is very difficult to interpret and see differences there. Crown closure should be a discrete variable (Table 1); here is instead plot as continuous. How would the story look like for the plot would be faceted by groups (~ crown closure)? Again, another dates are missing and it is unclear why only July 1st was selected? 

Thank you for your comment, we removed this figure from the text. 

• Fig. 9 = also Fir was considered as understory dominant species?? Missing from Tab .1 ? Unclear. Plot 9 is not necessary. 

Thank you for your comment, as described above in the GAM modelling only broadleaved/conifer understory share was taken as the predictor variable, but in the case of Scots pine it was possible to visualize differences in three different understory species (we added that information in the Table 1. ) 

Furthermore, we replaced this figure analogously to Fig. 4, showing understory in case of beech stands. Also here, we provided mean values only for the SWIR1 band during the year. However, if you still think it is not necessary we could remove it. 

• Fig, 12 Very difficult to interpret. Again, there are 4 crown closure groups, but only one smooth line, difficult to spot the differences. I suggest to remove. 

Thank you for your comment. We agree that this figure was hard to interpret and not very informative, therefore we removed it.

---

## [Decision Letter · Decision Letter 2]

1 Mar 2021

Evaluating the effect of stand properties and site conditions on the forest reflectance from Sentinel-2 time series

PONE-D-20-38051R2

Dear Dr. Grabska,

We’re pleased to inform you that your manuscript has been judged scientifically suitable for publication and will be formally accepted for publication once it meets all outstanding technical requirements.

Kind regards,

Michal Bosela, Ph.D.

Academic Editor

PLOS ONE

Additional Editor Comments (optional):

Reviewers' comments:

Reviewer's Responses to Questions

**Comments to the Author**

1. If the authors have adequately addressed your comments raised in a previous round of review and you feel that this manuscript is now acceptable for publication, you may indicate that here to bypass the “Comments to the Author” section, enter your conflict of interest statement in the “Confidential to Editor” section, and submit your "Accept" recommendation.

Reviewer #2: All comments have been addressed

2. Is the manuscript technically sound, and do the data support the conclusions?

Reviewer #2: Yes

3. Has the statistical analysis been performed appropriately and rigorously? 

Reviewer #2: Yes

4. Have the authors made all data underlying the findings in their manuscript fully available?

Reviewer #2: No

5. Is the manuscript presented in an intelligible fashion and written in standard English?

Reviewer #2: Yes

6. Review Comments to the Author

Reviewer #2: Well done, I have no further suggestions.

*Comment to Editor: In Data availability Statement, authors claim the full data availability. This is partially true, as all raw data are available as open access. Yet, processed datasets used to create individual models and plots are not directly shared. If PlosOne requires to share final datasets and code to reproduce the results, this could be shared using additional repository. Please consider what fits the best with PlosOne policy.

7. PLOS authors have the option to publish the peer review history of their article (what does this mean?). If published, this will include your full peer review and any attached files.

Reviewer #2: **Yes: **Mária Potterf

---

## [Editor Report · Acceptance letter]

5 Mar 2021

PONE-D-20-38051R2 

Evaluating the effect of stand properties and site conditions on the forest reflectance from Sentinel-2 time series 

Dear Dr. Grabska:

I'm pleased to inform you that your manuscript has been deemed suitable for publication in PLOS ONE. Congratulations! Your manuscript is now with our production department. 

Kind regards, 

on behalf of

Dr. Michal Bosela 

Academic Editor

PLOS ONE